# CASCADE: high-throughput characterization of regulatory complex binding altered by non-coding variants

## Graphical abstract

## Highlights

- CASCADE profiles binding of TF-COF complexes to DNA sequences

- CASCADE identifies cell-state-specific binding of complexes at CREs

- CASCADE can profile complexes affected by single-nucleotide polymorphisms

## Authors

David Bray, Heather Hook, Rose Zhao, ..., Leah C. Kottyan, Matthew T. Weirauch, Trevor Siggers

## Correspondence

tsiggers@bu.edu

## In brief

Bray et al. develop CASCADE, a method to profile transcription factor (TF)-cofactor (COF) complexes binding to DNA. They demonstrate the approach by profiling complex binding across the *CXCL10* cytokine promoter and to ∼1,700 single-nucleotide polymorphisms (SNPs). They anticipate that CASCADE can be applied to diverse biological systems to examine regulatory complex binding to DNA.

 Bray et al., 2022, Cell Genomics 2, 100098
February 9, 2022 © 2022 The Authors.

# Cell Genomics

CellPress

## Article

# CASCADE: high-throughput characterization of regulatory complex binding altered by non-coding variants

David Bray,[1,2,3,10] Heather Hook,[1,2,10] Rose Zhao,[1,2] Jessica L. Keenan,[1,2,3] Ashley Penvose,[1,2] Yemi Osayame,[1,2] Nima Mohaghegh,[1,2] Xiaoting Chen,[4] Sreeja Parameswaran,[4] Leah C. Kottyan,[4,7,8] Matthew T. Weirauch,[4,5,6,7] and Trevor Siggers[1,2,9,11,*]

[1]Department of Biology, Boston University, Boston, MA, USA
[2]Biological Design Center, Boston University, Boston, MA, USA
[3]Bioinformatics Program, Boston University, Boston, MA, USA
[4]Center for Autoimmune Genomics and Etiology, Cincinnati Children's Hospital Medical Center, Cincinnati, OH, USA
[5]Division of Developmental Biology, Cincinnati Children's Hospital Medical Center, Cincinnati, OH, USA
[6]Division of Biomedical Informatics, Cincinnati Children's Hospital Medical Center, Cincinnati, OH, USA
[7]Department of Pediatrics, University of Cincinnati College of Medicine, Cincinnati, OH, USA
[8]Division of Allergy and Immunology, Cincinnati Children's Hospital Medical Center, Cincinnati, Ohio 45229, USA
[9]Senior author
[10]These authors contributed equally
[11]Lead contact
*Correspondence: tsiggers@bu.edu

## SUMMARY

Non-coding DNA variants (NCVs) impact gene expression by altering binding sites for regulatory complexes. New high-throughput methods are needed to characterize the impact of NCVs on regulatory complexes. We developed CASCADE (Customizable Approach to Survey Complex Assembly at DNA Elements), an array-based high-throughput method to profile cofactor (COF) recruitment. CASCADE identifies DNA-bound transcription factor-cofactor (TF-COF) complexes in nuclear extracts and quantifies the impact of NCVs on their binding. We demonstrate CASCADE sensitivity in characterizing condition-specific recruitment of COFs p300 and RBBP5 (MLL subunit) to the *CXCL10* promoter in lipopolysaccharide (LPS)-stimulated human macrophages and quantify the impact of all possible NCVs. To demonstrate applicability to NCV screens, we profile TF-COF binding to ~1,700 single-nucleotide polymorphism quantitative trait loci (SNP-QTLs) in human macrophages and identify perturbed ETS domain-containing complexes. CASCADE will facilitate high-throughput testing of molecular mechanisms of NCVs for diverse biological applications.

## INTRODUCTION

Understanding how non-coding variants (NCVs) alter gene expression and lead to phenotypic differences remains an outstanding challenge in genomics. To date, genome-wide association studies (GWASs) have identified tens of thousands of NCV-trait associations, but the causal variants and their mechanism of action remain largely unknown.[1–5] A primary mechanism by which NCVs alter gene expression is by creating or disrupting binding sites for transcription factors (TFs) within cis-regulatory elements (CREs). Subsequently, a range of innovative experimental methods have been developed and used to study the impact of NCVs on TF-DNA binding; however, each possesses limitations to high-throughput annotation of NCV mechanisms. Traditional EMSAs (electrophoretic mobility shift assays)[6–8] and more recently developed mass-spectrometry-based approaches, such as PWAS (proteome-wide analysis of single-nucleotide polymorphisms [SNPs])[9] and FREP (flanking restriction enhanced pulldown),[10] have been used to examine allele-specific TF binding. However, these approaches allow only a few target NCVs to be analyzed at a time and are not ideally suited for high-throughput applications. Analyzing allelic imbalance in chromatin immunoprecipitation sequencing (ChIP-seq) data provides a powerful genome-scale approach to study the impact of NCVs on TFs and chromatin state,[4,11–16] but this approach is impractical for characterizing a target list of NCVs, as the chosen cell line must be heterozygous at all the target locations, which will not occur in most situations (e.g., when examining SNPs derived from population studies). To address the need for DNA-sequence flexibility, two recent high-throughput methods, both named SNP-seq,[10,17] were developed to screen DNA-oligo libraries for allele-specific binding of nuclear proteins via differential retention in a protein purification column[17] or via inhibiting restriction enzyme activity.[10] These SNP-seq methods can be used to analyze differential protein binding to hundreds of target NCVs but do not identify the proteins involved, which requires follow-up with lower-throughput experiments.

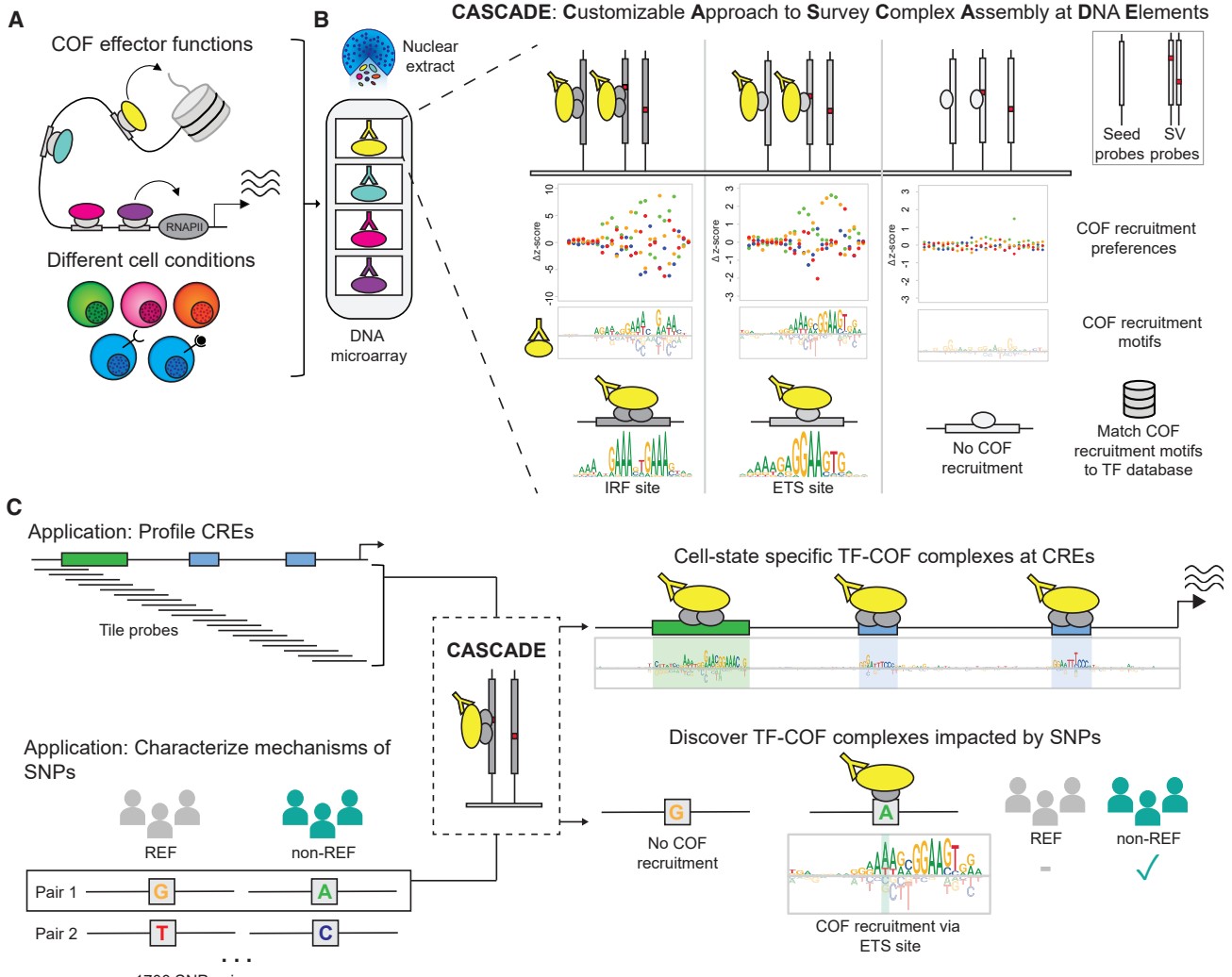

**Figure 1. Customizable approach to survey complex assembly at DNA elements (CASCADE) method and applications**

(A) Cofactors (COFs) affect transcription and chromatin state.

(B) COF recruitment to DNA is assayed using a DNA microarray on nuclear extracts from a cell type of interest. COF recruitment is assayed to a "seed" probe (e.g., genomic-derived TF binding site sequence) and all single variant (SV) probes. As shown in the confetti plots, COF recruitment to single variant probes yields nucleotide preferences along DNA sequence. Preferences are transformed to a COF recruitment motif (i.e., a COF recruitment logo). COF recruitment logos are matched to TF motif databases to infer TF identity.

(C) Overview of CASCADE applications. CASCADE can be applied to cis-regulatory elements (CREs) or to single-nucleotide polymorphism (SNP) pairs (reference [REF] and non-reference [non-REF] probes). Reference probes relate to the genomic consensus nucleotide sequence and non-reference to the phenotype-associated nucleotide variant. For CREs, tiling probes are used to span the genomic region and COF motifs for each tiling probe are integrated into a CRE-wide COF motif. For SNP pairs, COF recruitment motifs are determined for both and compared. IRF, interferon response factor; ETS, Erythroblast transformation specific.

Here we report the development of CASCADE (customizable approach to survey complex assembly at DNA elements), an array-based method used to identify DNA-bound TF-COF (cofactor) complexes and measure the impact of NCVs on their binding. CASCADE addresses the practical limitations of current methods, enabling high-throughput characterization of the impact of NCVs on gene regulatory complexes in a cell-specific manner. TFs primarily function by recruiting COFs to DNA, which subsequently alters gene expression through diverse mechanisms including interactions with the transcriptional machinery

(e.g., mediator), histone modification (e.g., EP300 histone acetylase), DNA modification (e.g., DNA methylases), chromatin remodeling (e.g., SWI/SNF-type complexes), or bridging to other COFs (e.g., BRD4) (Figure 1A).[18] CASCADE is used to profile COF recruitment to DNA microarrays with the goal of identifying functionally relevant TF-COF complexes and assessing the impact of NCVs on their DNA binding. We first use CASCADE to determine COF recruitment motifs that match established TF binding motifs, characterizing condition-specific recruitment of COFs p300 and RBBP5 (MLL subunit) to the *CXCL10*

promoter in lipopolysaccharide (LPS)-stimulated human macrophages. This demonstrates the accuracy of CASCADE for quantifying the impact of DNA variants on TF-COF binding. To demonstrate the utility of CASCADE in identifying TF-COF binding and the sequence specificity of COF recruitment, we profile TF-COF complex binding to CREs and non-coding SNP quantitative-trait loci (SNP-QTLs) in a stimulus-specific manner using human macrophages stimulated with LPS and interferon gamma (IFN-γ). CASCADE represents a methodological advance in the functional annotation of NCVs, establishing a direct link between NCVs and COFs that mediate diverse effects on gene regulation. We anticipate that CASCADE can be applied to a wide range of cellular systems for which antibody labeling reagents are available and sufficient high-concentration nuclear extracts can be generated, such as any cell line or primary cell or tissue system in which ~100–200 million cell nuclei can be acquired.

## RESULTS

### CASCADE
To identify cell-specific TF-COF complexes impacted by NCVs, we have developed the CASCADE method in which we use protein-binding microarrays (PBMs) incubated with nuclear extracts to profile the "recruitment" of COFs (e.g., EP300) to thousands of DNA probes (Figures 1A and 1B). This approach builds upon our prior work using nuclear extracts on PBMs to study TF-DNA binding in a cell-specific context.[19] This prior work demonstrated that PBMs can be used with total nuclear extracts instead of purified protein, enabling us to study how TF-DNA binding is affected by features such as cell-specific post-translation modifications and DNA-binding partners. In the current study, we demonstrate that by using nuclear extracts in our PBMs we can go a step further and label COFs instead of TFs, enabling us to study COF "recruitment" to DNA by TFs. As COFs interact broadly with many TFs,[20–22] profiling a single COF can report on many DNA-bound TF-COF complexes in a parallel manner without requiring knowledge of the TFs involved. Critically, by assaying COF recruitment to all single-nucleotide variants of a target DNA sequence, we can determine a COF "recruitment" motif. Comparison of the COF recruitment motif against TF motif databases allows us to annotate the recruiting TF to the level of TF family (Figure 1B). This method provides a cell-specific, *in vitro* approach to identify TF-COF complexes binding to CREs and to quantify the impact of NCVs on their binding.

### Application of CASCADE to CREs
To demonstrate the utility of CASCADE for profiling TF-COF binding and determining the DNA-sequence specificity of COF recruitment, we sought to examine a well-studied CRE with established TF binding sites bound by stimulus-dependent TF-COF complexes. We profiled the recruitment of the COF EP300, hereafter p300, to a promoter segment of the chemokine gene *CXCL10* in resting and LPS-stimulated human THP-1-derived macrophages. p300 is a broadly acting acetyltransferase that is recruited by diverse TFs, including both NF-κB and IRF3, which function at the *CXCL10* promoter.[20,23,24] The *CXCL10* promoter was chosen because it has been well studied and previously shown that the TFs that bind to this promoter are

stimulus dependent. In LPS-induced activation of *CXCL10* in macrophages, three separate TF binding sites in the promoter are required for full activation: two NF-κB binding sites and an interferon-sensitive response element (ISRE)[23,24] (Figure 2A), providing a test case for our CASCADE method.

To query p300 recruitment across the *CXCL10* promoter segment (166 bp), we assayed recruitment to 29 tiling probes (each 26 bp long) generated at 5 bp intervals across the target promoter region (Figure 1C; see STAR Methods, Table S1). For each tiling probe on our microarray, we also included all single variant (SV) probes to allow a COF recruitment motif to be determined every 5 bp (Figure 1C). A CRE-wide p300 recruitment motif was then generated for each experimental condition by integrating these individual motifs across their overlapping positions (Figure 2B, tracks 1–4; see STAR Methods).

Our CRE-wide recruitment motif identified that p300 recruitment to the three previously characterized TF binding sites occurred in an LPS-inducible manner (Figure 2B, tracks 1–4). These results are consistent with previous studies that demonstrated the LPS-inducible binding of IRF3 and NF-κB to the *CXCL10* promoter.[24–28] To infer the identity of the TFs involved, we compared the p300 recruitment motifs to a database of previously characterized TF binding motifs (see STAR Methods) and identified IRF3 and NF-κB as high-scoring matches (Figure S1A, track 1 and Figure S1B, track 2). To confirm the binding of NF-κB and IRF3 at these sites, we performed CASCADE experiments directly for the TFs RELA (the p65 subunit of NF-κB) and IRF3, using antibodies against the TFs instead of p300. p65 bound specifically to the previously characterized NF-κB sites and exhibited the expected DNA binding site specificity (Figure 2B, track 6 and Figure S2, track 14). IRF3 bound specifically to the ISRE[28,29] and weakly to the two NF-κB sites, which is consistent with the indirect tethering of IRF3 by NF-κB previously reported in LPS-stimulated macrophages (Figure 2B, track 5).[30,31] Critically, the binding motifs determined for IRF3 (Figure 2B, track 5) and p65 (Figure 2B, track 6) agree strongly with those for p300 (Figure 2B, tracks 1 and 2), demonstrating that CASCADE can quantify the impact of single-nucleotide variants on TF-COF binding with enough sensitivity to accurately capture the binding motifs for the underlying TF family.

To determine whether additional COFs with different effector functions are also recruited to the *CXCL10* promoter segment, we profiled the recruitment of RBBP5, a core subunit of the MLL histone lysine methyltransferase complex (Figure 2C; Table S1). Unlike the LPS-inducible recruitment of p300, RBBP5 is constitutively recruited to the *CXCL10* promoter sequences at comparable levels in the presence or absence of LPS (Figure 2D, tracks 7 and 8). RBBP5 is recruited only to the ISRE element and not the NF-κB sites, demonstrating a different recruitment preference than p300. However, as IRF3 binding to the ISRE is LPS-induced (Figure 2B, track 5 and Figure S2, track 13), our data suggest recruitment of RBBP5 to this site is independent of IRF3. Furthermore, the COF recruitment motifs for p300 and RBBP5 at the ISRE site exhibit clear differences in nucleotide preference (e.g., RBBP5 prefers a 5′-AAANCGAAA-3′ consensus whereas p300 prefers a 5′-GAACGGAAA-3′ consensus; Figure 2B, tracks 1 and 2 and Figure 2D, tracks 7 and 8). Comparing the RBBP5 recruitment motifs against a TF motif database

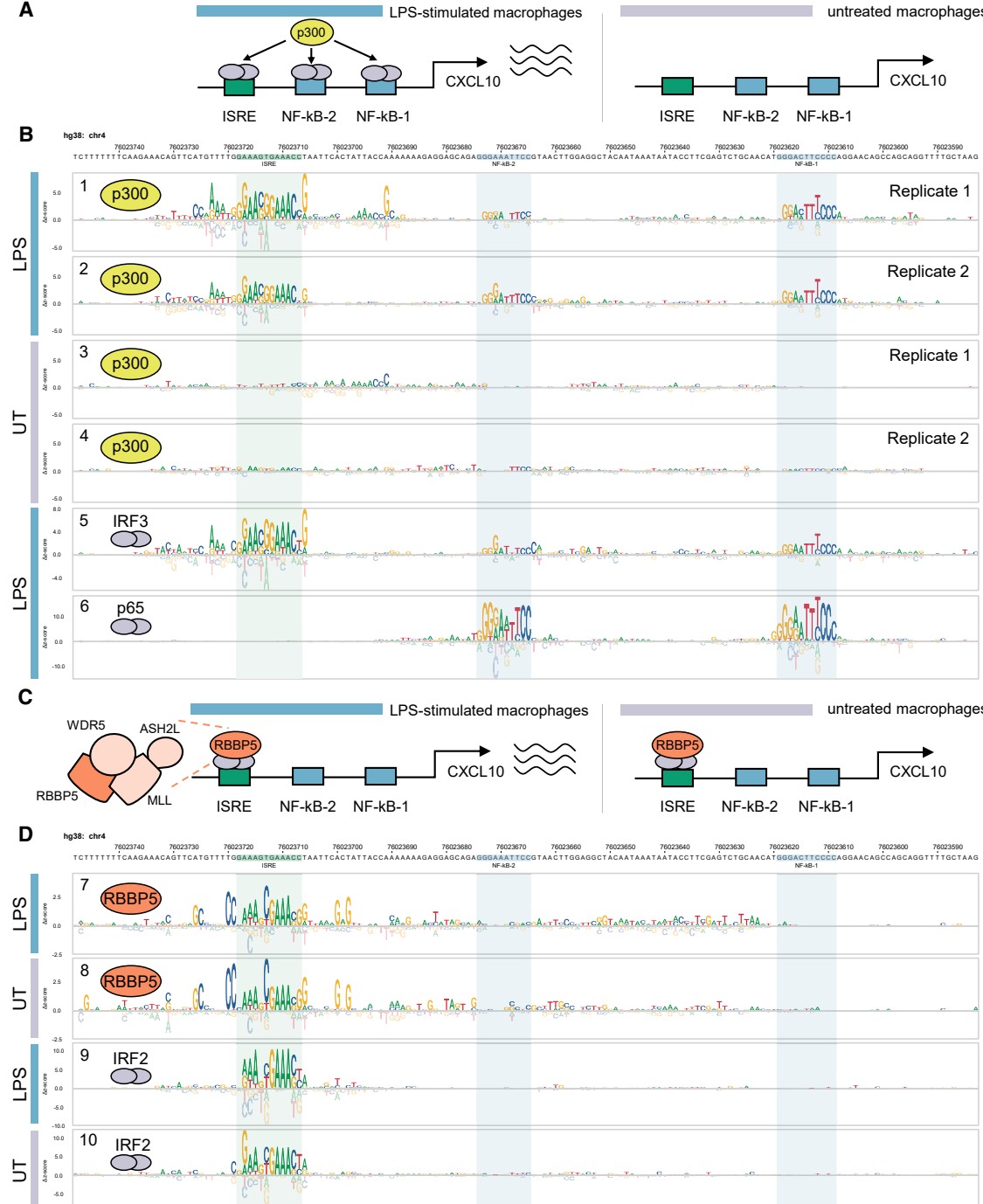

**Figure 2. CASCADE-based characterization of COF recruitment to the *CXCL10* promoter**

(A) Schematic of LPS-inducible recruitment of p300 to the *CXCL10* promoter in macrophages.

(B) CRE-wide p300 recruitment motif and TFs IRF3 and p65/RELA across the *CXCL10* promoter. Experiments using extracts from LPS-stimulated or untreated (UT) macrophages are indicated with colored bars. p300 recruitment motifs are shown for biological replicate experiments (Replicate 1 and 2).

(C) Schematic of condition-independent recruitment of RBBP5 to *CXCL10* promoter.

(D) CRE-wide motifs for COF RBBP5 and TF IRF2 across the *CXCL10* promoter segment. Experimental conditions as in (B). See also Table S1 and Figures S1, S2, and S7.

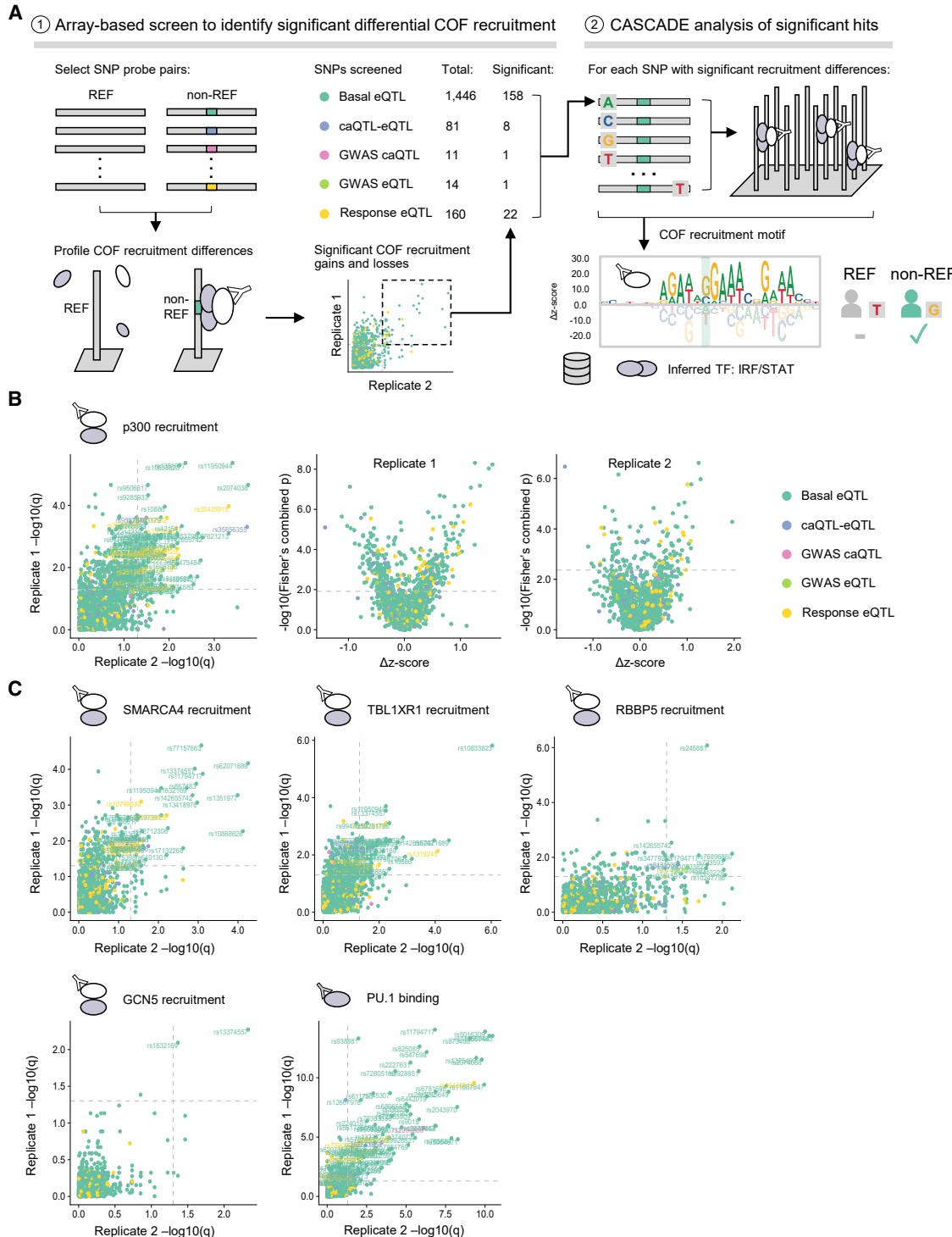

**Figure 3. CASCADE-based analysis of SNP-QTLs in human macrophages**

(A) Overview of two-step CASCADE-based approach to characterize 1,712 SNP-QTLs. (1) Step 1: screen for differential COF recruitment to SNP probe pairs by comparing recruitment to reference (REF) and non-reference (non-REF) alleles. The number of probe pairs in each QTL class for which significant COF recruitment was identified in at least one experiment is indicated. (2) Step 2: CASCADE-based motifs are generated for SNPs identified as significantly bound. COF motifs are compared against TF-motif databases to infer TF identity.

*(legend continued on next page)*

(see STAR Methods), we identified IRF2 as a high-scoring match (Figure S1C, track 7 and Figure S1D, track 8). IRF2, and the related IRF8, are both constitutively expressed in THP-1-derived macrophages, which would support the LPS-independent RBBP5 recruitment. CASCADE analysis of both IRF2 and IRF8 yielded CRE-wide motifs that closely matched those obtained for RBBP5 (Figure 2D, tracks 9 and 10 and Figure S2, tracks 11 and 12).

These results demonstrate several key features of CASCADE. First, this method can identify functionally relevant, stimulus-specific TF-COF complexes regulating *CXCL10,* demonstrating that it can be used to accurately identify DNA-bound TF-COF complexes binding to CREs. Second, our CRE-wide recruitment motifs, defined by an exhaustive analysis of all NCVs across the promoter, demonstrate the sensitivity of the approach to accurately quantify the impact of individual NCVs on TF-COF complexes and illustrate a powerful new approach to interrogate CREs and their regulatory inputs.

### Application of CASCADE to SNPs

To demonstrate the use of CASCADE to profile target lists of NCVs, we examined SNPs (i.e., single-base NCVs identified in population-level studies) associated with gene expression and chromatin state in human immune cells. To increase the number of SNPs that we could screen, we developed a hierarchical two-step approach (Figure 3A). In step one, COF recruitment to SNP probe pairs (reference and non-reference alleles) is screened to identify variants that lead to significant differential COF recruitment (Figure 3, step 1). In step two, to infer the identity of the TFs involved at each of these significant SNP loci, a second microarray is used to determine CASCADE-based COF recruitment motifs (Figure 3, step 2). The COF recruitment motifs generated for each SNP locus can then be compared to TF motif databases to infer the identity of the TF family and to provide additional context for assessing the impact of each SNP.

We used this two-step approach to profile COF recruitment to 1,712 SNPs that have been previously associated with gene expression (expression quantitative trait loci, eQTLs) and chromatin accessibility (caQTLs) changes in myeloid cells[3,32,33] (Figure 3A; Table S2). We performed our analysis with nuclear extracts from THP-1-derived macrophages stimulated with IFN-γ and LPS (see STAR Methods). To assess the impact of the SNP-QTLs on different cellular functions, we profiled recruitment of five COFs from different functional categories: p300, a histone acetyltransferase; SMARCA4/BRG1, a subunit of the SWI/SNF chromatin remodeling complex; TBL1XR1, a subunit of the nuclear receptor corepressor (NCoR) complex; RBBP5, a subunit of the MLL histone lysine methyltransferase complex; and GCN5, a histone acetyltransferase. In addition to these COFs, we screened for differential binding of the TF PU.1 due to its known role in establishing the myeloid enhancer landscape

and the previously demonstrated prevalence of the PU.1 binding motif at macrophage SNP-QTLs.[3,34,35]

Our step-one screen identified 164 total SNP-QTLs that reproducibly altered the recruitment of at least one of the tested COFs (Figures 3B and 3C), representing 9.6% of the sites examined. Except for the GWAS caQTL category, comparable proportions of the SNP-QTL categories tested reproducibly altered COF recruitment: 136 basal eQTLs (9.4%), 7 caQTL-eQTLs (8.6%), 1 GWAS eQTL (7.1%), and 20 response eQTLs (12.5%). Profiling the TF PU.1, we also observed widespread differential PU.1 binding at 95 SNP-QTLs (Figure 3C), including 23 that coincided with the differential recruitment of at least one of the COFs screened.

By examining the direction of the differential recruitment, we identified SNP-QTLs that caused gain or loss of TF-COF binding (Figures 3B and S3). For example, our screen identified 63 non-reference alleles that led to statistically significant gain of p300 recruitment (Figure 3B, rightmost two panels, positive $\Delta Z$ score) and non-reference alleles that led to a significant loss relative to the reference allele (Figure 3B, rightmost two panels, negative $\Delta Z$ score). In total, across all COFs and TFs screened, we observed differential recruitment and/or binding at 243 of the 1,712 non-reference (14.2%) alleles with 134 gains, 108 losses, and one locus demonstrating both. Of note, for each SNP-QTL that exhibited significant differential recruitment of more than one COF (40 total), the direction of the effect, either gain or loss, was consistent across each COF.

To compare the results of our SNP-QTL screen with existing approaches to characterize COF recruitment at NCVs, we performed allelic imbalance analysis on 436 publicly available ChIP-seq experiments for the THP-1 cell line (Tables S3 and S4). In brief, this analysis uses the MARIO pipeline[36] to examine sequencing read imbalance between the two alleles of a given heterozygous genetic variant in the given experiment. Of the 1,712 SNP-QTLs we screened on our array-based platform, only 305 (17.8%) were heterozygous in THP-1 cells (Figure S4A), highlighting the limitation of genomic assays such as ChIP-seq to characterize genetic variants that may not occur naturally in a given cell line but that do occur naturally within a population. Of the 28 SNP-QTLs that demonstrated reproducible differential COF recruitment in our screen and are heterozygous in THP-1 cells, 9 (32.1%) were found to also have allelic effects on at least one general chromatin-associated feature assayed through ChIP-seq (e.g., COFs, histone marks, CTCF, RNAP subunits; see STAR Methods) (Figure S4A). Except for PU.1, which was found to have allele-specific binding at four of these SNP-QTLs (Figure S4A; discussed further below), expanding our analysis to all TFs did not identify allelic imbalance at any more of these 28 loci beyond the nine already associated with general chromatin features. These results demonstrate that our COF-centered approach can be used to screen broad classes of SNP-QTLs for both gains and losses of TF-COF complex

---

(B) Comparison of p300 differential recruitment across biological replicates. Comparison of q-values for replicates is shown (left). The q-values represent the statistical significance of the difference between p300 recruitment measured at REF and non-REF probes within a SNP pair adjusted for multiple hypothesis testing (see STAR Methods). Comparison of statistical significance (q-values) against the difference in p300 recruitment Z score between REF and non-REF probes in each SNP pair in technical replicates 1 (middle) and 2 (right) is shown. QTL class for each SNP is indicated.

(C) Comparison of differential COF recruitment across biological replicates is shown for candidate COFs and the TF PU.1. eQTL, expression quantitative trait loci; caQTL, chromatin accessibility quantitative trait loci. See also Table S1 and Figures S3, S4, S5, S6, and S7.

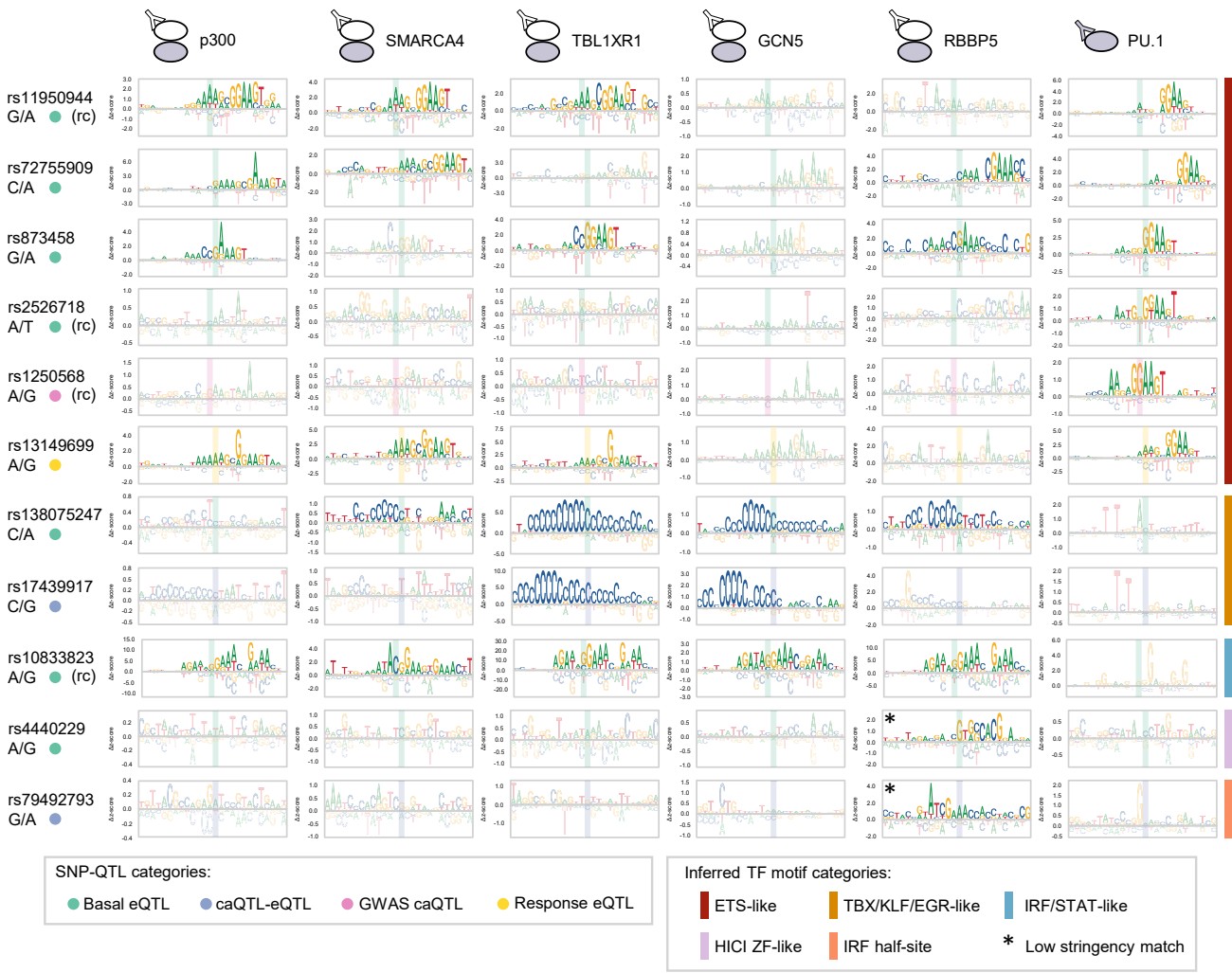

**Figure 4. CASCADE-determined motifs at SNP loci**
COF recruitment motifs for p300, SMARCA4, TBL1XR1, GCN5, and RBBP5 are shown for 10 SNP-QTL loci. PU.1 binding motifs at each locus are also shown. Position of the SNP location within each motif is shown with a shaded rectangle. (*) denotes a motif match obtained at lower stringency (see STAR Methods). QTL type of each SNP is indicated (left-hand side, colored dots). Only sites that met an imposed seed Z score threshold were plotted (see STAR Methods). Corresponding reference and SNP are shown beneath each rsID. (rc) denotes a site plotted as its reverse complement relative to the reference strand. For these sites, the reference and non-REF alleles are also indicated as their complementary nucleotides. See also Table S1 and Figures S4, S5, S6, and S7.

binding with *in vivo* relevance. Furthermore, the ability to use customizable DNA probes to profile any possible NCV represents a key advantage over ChIP-based approaches that are constrained to heterozygous variants available in cell lines.

For step two of our SNP-QTL analysis, we used CASCADE to determine COF recruitment motifs at select loci. These motifs allow us to infer the identity of the TFs mediating differential COF recruitment at each locus (Figure 3A, step 2). We selected 158 basal eQTLs, 8 caQTL-eQTLs, 1 GWAS caQTL, 1 GWAS eQTL, and 22 response eQTLs, as these loci showed significant differential recruitment of one or more of the regulators screened (see STAR Methods and Table S5). To determine our COF recruitment motifs, we profiled the base preferences of the local genomic region (26 bp) centered at each of these SNP-QTLs. Consistent with our observed differential PU.1 binding, the

COF recruitment motifs for many loci matched ETS-family binding motifs (GGAA core) (Figure 4). COF motifs were also identified that matched TBX/KLF/EGR-family zinc finger motifs and IRF/STAT-family motifs, and two motifs only matched a known motif at a relaxed stringency (Figure 4; see STAR Methods). Comparing the recruitment motifs generated at a given SNP-QTL locus, we found the motif base preferences and alignment were consistent across COF and PU.1 experiments, confirming a common underlying TF-COF complex. Examining SNP-QTLs specifically affecting ETS motifs, we found that variants can impact different positions along the ETS motif, including both the variable 5′ flanking region (rs11940944, rs72755909, rs2526718) and the core ETS 5′-GGAA-3′ element (rs873458, rs1250568). The CASCADE-determined COF motifs for SNP-QTL at rs72755909 matched an ETS-family motif for p300,

SMARCA4, RBBP5, and PU.1. In our allelic imbalance analyses, this locus is the only example of a heterozygous allele that was found to perturb the binding of PU.1 as well as a chromatin feature (Figures S4A and S4B, red). Together, these results highlight that COF recruitment motifs can provide a means to understand the molecular mechanisms underlying a SNP-QTL.

Our motif analyses also identified that a number of COFs share similar motif profiles, suggesting that at these loci they are recruited by a common TF or by related TF-family members. We first examined whether this is a result of the particular QTLs analyzed or a more general feature of these COFs. To examine COF specificity in our CASCADE assay, we designed a new microarray to profile COF recruitment to a panel of ~350 consensus TF sites (and single-nucleotide variants) derived from motifs in the JASPAR database (see STAR Methods). Profiling COF recruitment to this diverse panel of sites identified common recruitment to some TF families as seen for the QTL loci (e.g., ETS and IRF families) but also identified COF-specific recruitment preferences (Figure S5). For example, only TBL1XR1 was strongly recruited to nuclear-receptor-family sites, and only RBBP5 was strongly recruited to POU-family sites (Figure S5). To determine whether common recruitment preferences might be a result of "bridging factors" known to mediate interactions between COFs and TFs, we also profiled recruitment of mediator subunit MED1 and the scaffold protein BRD4. However, we found that these COFs exhibited their own distinct profiles, suggesting that mediator or other scaffold proteins do not likely explain common COF recruitment preferences. These results suggest that recruitment to common TFs is not a general feature of our assay but is an inherent feature of these COFs and the loci examined. Next, to examine COF recruitment specificity *in vivo*, we examined available ENCODE ChIP-seq datasets where four of the COFs in our study (EP300, SMARCA4, TBL1XR1, and RBBP5) were profiled in K562 cells.[37] We found that these COFs exhibited highly overlapping binding profiles (Figure S6). For example, of the 24,744 EP300 binding loci, 83% overlapped with binding of these other three COFs. Further, 38% of EP300 binding overlapped with both SMARCA4 and TBL1XR1, the two COFs that exhibited similar motif profiles in our analyses. Together, these results suggest that similar COF recruitment profiles seen in our analysis are a result of direct recruitment of these COFs by individual TFs or TF-family members and that this is consistent with overlapping recruitment profiles seen *in vivo*.

Finally, we highlight two specific gain-of-recruitment SNP-eQTLs identified in our screen to demonstrate how CASCADE can be used to generate mechanistic models of NCVs. Our analysis for rs11950944 (G/A), a basal SNP-eQTL in myeloid cells,[5] found that p300 (Z score: 2.36), SMARCA4 (Z score: 2.99), and TBL1XR1 (Z score: 2.61) are recruited to the non-reference allele but are either not recruited or are below our detection threshold for the reference allele (p300: Z score: −0.13, SMARCA4: Z score: −0.38, TBL1XR1: Z score: 0.37) (Figure 5A left; Table S5). The COF recruitment motifs for all three COFs matched with ETS-factor motifs (Figure 5A, right). Consistent with our motif-based inferences, the ETS factor PU.1 preferentially bound the non-reference allele (Z score: 5.99), though it could also be detected at the reference allele (Z score: 4.04). These results suggest a model where the non-reference allele enhances the

DNA binding of an ETS-family TF, possibly PU.1, which leads to enhanced recruitment of these COFs (Figure 5C). We note that enhanced binding of PU.1 at DNA variants in murine myeloid cells has been previously shown to correlate with increased local histone modifications characteristic of primed and active regulatory elements as well as with increased transcriptional output.[35]

Our analysis for a second basal SNP-eQTL, rs10833823 (A/G), in myeloid cells[5] identified a different scenario in which the entire panel of COFs tested were recruited to the reference allele, but the non-reference allele caused significantly higher recruitment for three of the COFs: TBL1XR1 (Z scores: REF = 9.71, non-REF = 28.49), GCN5 (Z scores: REF = 1.54, non-REF = 3.36), and RBBP5 (Z scores: REF = 10.56, non-REF = 15.82) (Figure 5B left; Table S5). The COF recruitment motifs for all COFs matched GA-rich IRF/STAT-family motifs (Figure 5B, right), and consistent with our inference of recruitment by IRF/STAT-type TFs, we did not observe PU.1 binding at this site (Figure 5B, left). Notably, while the non-reference allele enhanced COF recruitment in our assay, it occurred at a low-information position in the IRF/STAT motifs that did not appreciably affect the position weight matrix (PWM) scores for these TFs. Therefore, a computational screen for this SNP-QTL would not predict appreciable changes in TF binding and would be missed. Our results for this SNP-QTL suggest that the non-reference allele does not affect TF binding but can alter the recruitment of COFs (Figure 5D), possibly by a mechanism involving DNA-based allostery.[38,39] These results demonstrate how the CASCADE approach, based on COF recruitment profiling, can generate mechanistic models for how NCVs can alter the binding of TF-based regulatory complexes.

## DISCUSSION

CASCADE addresses the current need for high-throughput methods to determine the impact of NCVs on TF-COF regulatory complex binding. Several features highlight the applicability of the approach and distinguish it from existing methods. First, by assaying recruitment of broadly acting COFs and inferring TFs via COF recruitment motifs, CASCADE provides the first high-throughput method to both discover and characterize TF-COF complexes impacted by NCVs. By performing our screen with a panel of broadly acting COFs from diverse functional categories, we can profile a large fraction of potentially functionally relevant TF-COF complexes. Our analyses of the *CXCL10* promoter (Figure 2), SNP loci (Figure 4), and an expanded set of synthetic loci (Figure S6) indicate that a range of different TF families are being identified, despite using only a small number of COFs. However, it remains unclear how many of the ~1,600 human TFs we can profile using this approach. Further studies using expanded sets of COFs will help to clarify how many TFs can be profiled using CASCADE and establish whether some TFs are more efficiently profiled via COFs than others.

Second, the CASCADE method profiles TF-COF complexes, rather than TFs alone, allowing us to link NCVs with the diverse biological functions mediated by COFs, such as histone modifications and chromatin remodeling. This provides an additional level of biological annotation beyond current approaches that

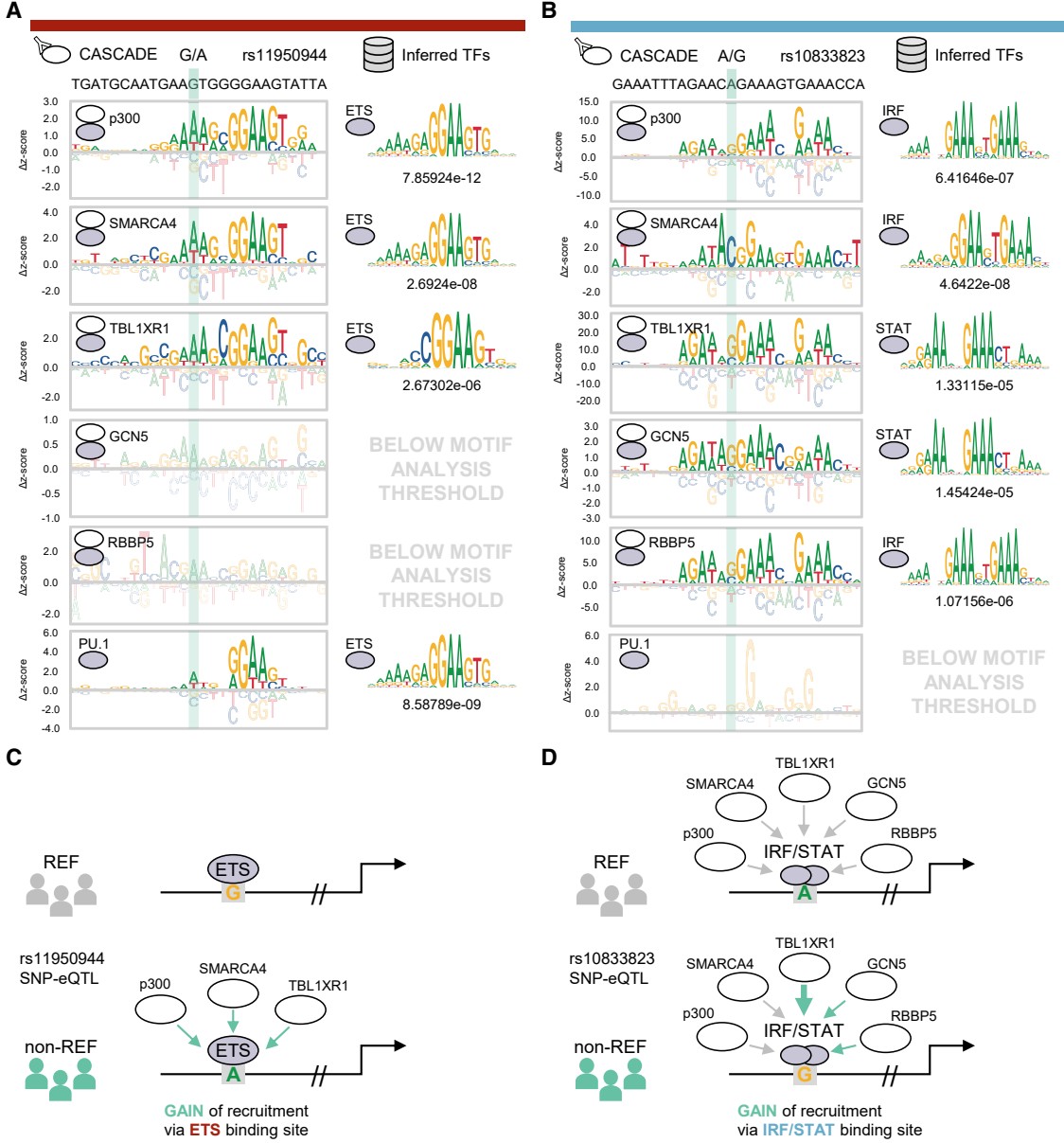

**Figure 5. Constructing models with CASCADE for SNP-eQTLs**
(A) Left column: CASCADE-determined COF recruitment motifs for p300, SMARCA4, TBL1XR1, GCN5, and RBBP5 at the local genomic region surrounding rs11950944. PU.1 binding motif is also shown. Right column: TF binding motif with the strongest association to each corresponding CASCADE COF recruitment motif. Statistical significance (p value) for TF matching is shown below each TF motif (see STAR Methods). Position of the SNP location within each motif is shown in the shaded area. QTL type and inferred TF category are indicated by the same color scheme as in Figure 4.
(B) Same as in (A) but for the local genomic region surrounding rs10833823. Only sites that met an imposed Z score threshold were plotted and used for motif analysis (see STAR Methods).
(C) Integrative model for COF recruitment changes at SNP-eQTL rs11950944.
(D) Same as in (C) but for SNP-eQTL rs10833823. See also Table S1 and Figures S5, S6, and S7.

focus on TF binding alone (e.g., MITOMI, SMiLE-seq, CSI, PBM, SELEX-seq, nextPBM, etc.[19,40–48]). Finally, by combining the flexibility of the customizable, array-based approach with the use of nuclear extracts, CASCADE can be used to analyze tailored lists of target NCVs in a cell-state-specific manner. This sequence flexibility will allow for the analysis of diverse

types of NCVs, including somatic variants in cancer, rare variants found in population studies, or even synthetic variants as analyzed in our *CXCL10* promoter analysis. Analyzing such NCVs is currently impractical with standard ChIP-seq-based approaches, which are confined to the genotypes of available cells.

## Limitations of study

We anticipate the CASCADE method to be broadly applicable to a range of biological systems for which antibody labeling reagents are available and high-concentration nuclear extracts can be generated, such as any cell line or primary cell or tissue system in which ~100–200 million cell nuclei can be acquired. However, there are several additional practical requirements and issues to be considered. The first is how many loci can be analyzed, either CREs or SNP loci. Using the Agilent 4x180K microarray platform discussed here, we can profile 1,091 bp of CRE sequence in a single experiment at the single-nucleotide level. However, to increase coverage in our SNP screen, we described a two-stage approach in which we first screened binding to pairs of REF/non-REF loci (up to 8,467 can be screened) and then assayed a reduced set of significant loci at the single-nucleotide level to identify COF motifs (up to 214 loci can be analyzed at this level). A similar two-stage approach could also be used to more efficiently profile TF-COF binding to CREs to achieve greater coverage. By first profiling binding using 5-bp overlapping tile sequences, we could screen 84,691 bp of CRE sequence from the genome and then assay 214 significantly bound loci at the single-nucleotide level.

Second, there is the question of how our binding results relate to genomic binding of regulatory complexes when chromatin factors (e.g., histones) are not present on our DNA arrays. As with all *in vitro* assays, our approach is meant to characterize the possible complexes that can assemble on each DNA loci when DNA is accessible. While these loci may be inaccessible in the native chromatin environment, one can integrate this approach with chromatin maps to bias the loci being analyzed. For example, one might first identify open chromatin regions using ATAC-seq or by profiling histone marks and then follow up with a CASCADE analysis to profile the TF-COF binding and impact of SNPs.

The third is the technical requirement of CASCADE for high-concentration nuclear extracts and antibodies in order to probe the COFs of interest, which may be difficult to obtain when cellular material is limited and antibodies are not available. The fourth is the need for additional analyses to refine TF annotations beyond those attainable by motif comparison. In our approach, DNA-bound TFs are identified indirectly via motif comparison against COF recruitment motifs. Therefore, our approach is unable to distinguish between related TFs that have similar DNA binding motifs, which is often the case for members of the same TF family. Integrating CASCADE results with other data types, such as gene expression or protein abundance data, can provide a way to refine our TF annotations for follow-up experiments, as was done for the IRF and NF-κB-family proteins in our *CXCL10* promoter analysis. Finally, it remains unclear the extent to which multi-protein COF complexes are assembled on our DNA microarrays in their native conformation and whether we are assaying the recruitment of smaller sub-complexes or even single COFs (i.e., binary TF-COF interactions). In either case, we demonstrate that profiling subunits of multi-protein COF complexes (e.g., the RBBP5 subunit of MLL complexes) can sensitively and accurately report on the impact of NCVs on the binding of associated TFs, validating the basic approach.

## Conclusions

To demonstrate the utility of the CASCADE method, we applied it to the functional characterization of CREs and SNP-QTLs in resting and LPS-stimulated human macrophages. By profiling recruitment of activators (p300 and RBBP5) to the *CXCL10* promoter, we identified binding of stimulus-specific TF-COF complexes and exhaustively mapped the impact of all NCVs on TF-COF binding across the promoter region. We propose that this approach can be used to map any CRE to uncover cell- or stimulus-specific TF-COF regulatory complexes and their DNA-sequence dependence. Using CASCADE to profile 1,712 SNP-QTLs previously associated with chromatin or gene expression changes, we identified altered binding for a number of ETS-family TF-COF complexes. We anticipate that CASCADE can be used to characterize the molecular impact of NCVs across diverse cell and stimulation conditions. Important next steps will be to apply the CASCADE approach to a range of disease-associated SNPs and NCVs using both cell lines and primary human cells to provide functional annotations and insights about the mechanisms of DNA variants.

## STAR★METHODS

Detailed methods are provided in the online version of this paper and include the following:

- KEY RESOURCES TABLE
- RESOURCE AVAILABILITY
  - Lead contact
  - Materials availability
  - Data and code availability
- EXPERIMENTAL MODEL AND SUBJECT DETAILS
  - Cell lines
- METHOD DETAILS
  - CASCADE method
  - Cell culture
  - CASCADE PBM experimental methods
  - CASCADE microarray designs and analyses
  - CASCADE microarray designs and analyses for TFBS
  - Motif similarity analysis
  - Identification and processing of THP-1 ChIP-seq data
  - Allele-dependent ChIP-seq data analysis
  - ENCODE ChIP-seq data analysis
- QUANTIFICATION AND STATISTICAL ANALYSIS
  - CASCADE analyses for genomic loci
  - Determination of SNPs to include for CASCADE
  - Motif similarity analysis

### SUPPLEMENTAL INFORMATION

### ACKNOWLEDGMENTS

This work is supported by National Institutes of Health (NIH) R01 A116829, NIH R21 HG011289, and NIH RO1 AI51051 to T.S. and NIH R01 NS099068, R01 AR073228, R01 HD010730, R01 AI024717, U01 AI130830 and R01 GM055479 to M.T.W. We thank Juan Fuxman Bass, Thomas Gilmore, Andrew Emili, and David J. Waxman for discussions related to this work.

## AUTHOR CONTRIBUTIONS

D.B., H.H., and T.S. jointly conceived of the study. D.B. developed the software to design, analyze, and visualize CASCADE experiments. H.H. performed the experimental work. J.L.K., A.P., N.M., R.Z., and Y.O. supported H.H. with validation of the experimental technique. D.B., H.H., and T.S. analyzed and interpreted the experimental results. S.P., X.C., L.C.K., and M.T.W. performed allelic ChIP-seq analyses. T.S. oversaw the experimental work and development of the computational methods. D.B., H.H., and T.S. wrote the manuscript.

## DECLARATION OF INTERESTS

The authors declare no competing financial interests. David Bray is currently employed at Foundation Medicine Inc. (Cambridge, MA, USA). Ashley Penvose is currently employed at Generation Bio (Cambridge, MA, USA). Jessica L. Keenan is currently employed at Generation Bio (Cambridge, MA, USA). Yemi Osayame is currently employed at Takeda Pharmaceuticals (Cambridge, MA, USA). Nima Mohaghegh is currently employed at 4D Molecular Therapeutics (San Fransisco, CA, USA).

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

## STAR★METHODS

### KEY RESOURCES TABLE

| REAGENT or RESOURCE | SOURCE | IDENTIFIER |
|---|---|---|
| **Antibodies** | | |
| Anti-p300 | Aabcam | Cat#ab14984; RRID: AB_301550 |
| Anti-Brg-1 | Santa Cruz Biotechnology | Cat#sc17796; RRID: AB_626762 |
| Anti-Gcn5 | Santa Cruz Biotechnology | Cat#sc365321x; RRID: AB_10846182 |
| Anti-RBBP5 | Bethyl Laboratories | Cat#A300-109A; RRID: AB_210551 |
| Anti-TBL1XR1 | Santa Cruz Biotechnology | Cat#sc100908; RRID: AB_1130006 |
| Anti-HDAC1 | Abcam | Cat#ab7028; RRID: AB_305705 |
| Anti-MED1 | Bethyl Laboratories | Cat#A300-793A; RRID: AB_577241 |
| Anti-BRD4 | Bethyl Laboratories | Cat#A301-985A50; RRID: AB_1576498 |
| Anti-NFkB p65 | Santa Cruz Biotechnology | Cat#sc372; RRID: AB_632037 |
| Anti-NFkB p65 | Santa Cruz Biotechnology | Cat#sc8008; RRID: AB_628017 |
| Anti-ICSBP | Santa Cruz Biotechnology | Cat#sc6058x; RRID: AB_649510 |
| Anti-IRF3 | Cell Signaling Technology | Cat#D83B9; RRID: AB_1904036 |
| Anti-IRF2 | Santa Cruz Biotechnology | Cat#sc374327; RRID: AB_10990717 |
| Anti-PU.1 | Santa Cruz Biotechnology | Cat#sc352; RRID: AB_632289 |
| Donkey anti-Goat IgG (H+L) Cross-Adsorbed Secondary Antibody, Alexa Fluor 488 | ThermoFisher Scientific | Cat#A-11055; RRID: AB_2534102 |
| Goat anti-Mouse IgG (H+L) Highly Cross-Adsorbed Secondary Antibody, Alexa Fluor 488 | ThermoFisher Scientific | Cat#A-11029; RRID: AB_138404 |
| Goat anti-Rabbit IgG (H+L) Highly Cross-Adsorbed Secondary Antibody, Alexa Fluor 488 | ThermoFisher Scientific | Cat#A-11034; RRID: AB_2576217 |
| Goat anti-Mouse IgG (H+L) Highly Cross-Adsorbed Secondary Antibody, Alexa Fluor Plus 647 | ThermoFisher Scientific | Cat#A32728; RRID: AB_2633277 |
| Goat anti-Rabbit IgG (H+L) Highly Cross-Adsorbed Secondary Antibody, Alexa Fluor Plus 647 | ThermoFisher Scientific | Cat#A32733; RRID: AB_2633282 |
| Goat anti-Rabbit IgG (H+L) Cross-Adsorbed Secondary Antibody, HRP | ThermoFisher Scientific | Cat#G-21234; RRID: AB_2536530 |
| Goat anti-Mouse IgG (H+L) Cross-Adsorbed Secondary Antibody, HRP | ThermoFisher Scientific | Cat#G-21040; RRID: AB_2536527 |
| **Chemicals, peptides, and recombinant proteins** | | |
| Lipopolysaccharides from *Escherichia coli* O111:B4 | Sigma-Aldrich | Cat#L3024 |
| Human IFN-gamma Recombinant Protein | ThermoFisher Scientific | Cat#PHC4031 |
| Protease Inhibitor Cocktail | Sigma-Aldrich | Cat#P8340 |
| Phosphatase Inhibitor Cocktail A | Santa Cruz Biotechnology | Cat#sc45055 |
| Phorbol 12-myristate 13-acetate | Sigma-Aldrich | Cat#P8139 |
| **Deposited data** | | |
| RBBP5 ChIP-seq data, performed in K562 cells | ENCODE | ENCFF666PCE |
| EP300 ChIP-seq data, performed in K562 cells | ENCODE | ENCFF755HCK |

*(Continued on next page)*

***Continued***

| REAGENT or RESOURCE | SOURCE | IDENTIFIER |
|---|---|---|
| SMARCA4 ChIP-seq data, performed in K562 cells | ENCODE | ENCFF267OGF |
| TBL1XR1 ChIP-seq data, performed in K562 cells | ENCODE | ENCFF868SWL |
| CASCADE microarray data | GEO | GSE148945 |
| Experimental models: Cell lines | | |
| THP-1 | ATCC | Cat#TIB-202 |
| Oligonucleotides | | |
| Protein Binding Microarray Double-stranding Primer | Eurofins | 5′ - CAGCAGCGCTCAAGG AATCAAGAC - 3′ |
| Software and algorithms | | |
| RColorBrewer, version 1.1.2 | Neuwirth[49] | N/A |
| Cowplot, version 1.1.1 | Wilke[50] | N/A |
| ggseqlogo R package, version 0.1 | Wagih[51] | N/A |
| TOMTOM (MEME suit), version 5.0.3 | Patwardhan et al.[52] | N/A |
| FASTQC, version 0.11.8 | Kalita et al.[53] | N/A |
| Trim Galore, version 0.4.2 | Goodwin et al.[54] | N/A |
| cutadapt, version 1.9.1 | Bentley et al.[55] | N/A |
| bowtie2, version 2.3.4.1 | Langmead and Salzberg[56] | N/A |
| samtools, version 1.8.0 | Patwardhan et al.[52] | N/A |
| IMPUTE2, version 2.3.0 | Verma et al.[36] | N/A |
| cnvPartition, version 3.2.1 | Illumina Genome Studio | N/A |
| MARIO, version 3.9.3 | Verma et al.[36] | N/A |
| UpSetR Package, version 1.4.0 | Conway et al.[57] | N/A |
| CASCADE R-scripts, version 1.0.0 | https://github.com/Siggers-Lab/CASCADE_paper | N/A |
| ggplot2, version 3.3.5 | Wickham[58] | N/A |

## RESOURCE AVAILABILITY

### Lead contact
Further information and requests for resources and reagents should be directed to and will be fulfilled by the lead contact, Trevor Siggers (tsiggers@bu.edu).

### Materials availability
This study did not generate new unique reagents.

### Data and code availability

- Results of all CASCADE array experiments performed here have been deposited in the Gene Expression Omnibus and are publicly available (GEO accession: GSE148945). THP-1 genotyping array data are publicly available at dbGaP under accession number phs001989.v1. This paper analyzes existing, publicly available data. Accession numbers for the datasets are listed in the key resources table.
- All original code has been deposited on on Github (https://github.com/Siggers-Lab/CASCADE_paper) and is publicly available.
- Additional information required to reanalyze the data reported in this paper is available from the lead contact upon request.

## EXPERIMENTAL MODEL AND SUBJECT DETAILS

### Cell lines
THP-1 cells, a human monocyte cell line, were obtained and verified by ATCC. This cell line (ATCC, TIB-202) was established from a make who had acture monocytic leukemia.

## METHOD DETAILS

### CASCADE method

The CASCADE method is used to profile the impact of non-coding DNA variants on TF-COF complex binding and specifically refers to PBM experiments performed using cell nuclear extracts in which COFs, as opposed to TFs, are fluorescently labeled in the experiment. Experimental details are included below in the section titled 'CASCADE PBM experimental methods.' Applied to DNA probes that differ by single nucleotides, this method can be used to quantify the impact of SNPs on TF-COF binding. Applied to an exhaustive set of single-nucleotide variants across a DNA locus the method can be used to infer the identity of the TF recruiting the target COF to that DNA site. Details of these applications of CASCADE, including microarray design and data analysis, are described below in sections titled 'CASCADE microarray designs and analyses' and 'Motif similarity analysis.' A specific application used in this work to profile TF-COF binding to known TF binding sites is further outlined in the section 'CASCADE microarray designs and analyses for TFBS.'

### Cell culture

THP-1 cells, a human monocyte cell line, were obtained from ATCC (TIB-202). The cells were grown in suspension in RPMI 1640 Glutamax media (Thermofisher Scientific, Catalogue #72400120) with 10% heat-inactivated fetal bovine serum (Thermofisher Scientific, Catalogue #11360070) and 1mM sodium pyruvate (Thermofisher Scientific, Catalogue #16140071). T175 (Thermofisher Scientific, Catalogue #132903) non-treated flasks were used when culturing THP-1 cells for experiments. Cells were grown in 50mL of media when being cultured in T175 flasks.

To differentiate THP-1 cells into adherent macrophages, cells were grown to a density of $8.0 \times 10^5$ cells/mL and treated with 25ng/mL Phorbol 12-Myristate 13-Acetate (PMA) (Sigma-Aldrich, Catalogue #P8139) for 4 days. Following the 4 days of PMA treatment, the media was replaced with fresh RPMI media with 10% heat-inactivated fetal bovine serum and 1mM sodium pyruvate. The cells rested for two days in the fresh media before being stimulated with various reagents.

THP-1 cells differentiated with PMA were treated with either Lipopolysaccharide (LPS) (Sigma-Aldrich, L3024) or Interferon gamma (IFN-$\gamma$) (Thermofisher Scientific, Catalogue #PHC4031) in combination with LPS. PMA treated THP-1 cells were treated with 1$\mu$g/mL of LPS for 45 min or with 100ng/mL IFN-$\gamma$ for 2 h followed by 1$\mu$g/mL LPS for 1 h. For each condition, nuclear lysates were harvested. For all nuclear lysates assayed using PBM experiments, the expression levels of COFs and TFs profiled with CASCADE were confirmed by western blotting (Figure S7).

### CASCADE PBM experimental methods

The nuclear extract protocols are as previously described,[19] with modifications as detailed below in this section. To harvest nuclear extracts from THP-1 cells, the media was aspirated off and the cells were washed once with 1X PBS (Thermofisher Scientific, Catalogue #100010049). Once the 1X PBS used to wash the cells was aspirated off, enough 1X PBS was mixed with 0.1mM Protease Inhibitor (Sigma-Aldrich, Catalogue #P8340) to cover the cells was added to each flask. A cell scraper was used to dislodge the cells from the flask, and cells were collected in a falcon tube and placed on ice. Cells were pelleted by centrifugation at 500xg for 5 min at 4°C, and the supernatant was aspirated off. To lyse the plasma membrane, the cells were resuspended in Buffer A and incubated for 10 min on ice (10mM HEPES, pH 7.9, 1.5mM MgCl, 10mM KCl, 0.1mM Protease Inhibitor, Phosphatase Inhibitor (Santa-Cruz Biotechnology, Catalogue #sc-45044), 0.5mM DTT (Sigma-Aldrich, Catalogue #4315)). After the 10 min incubation, Igepal detergent (final concentration of 0.1%) was added to the cell and Buffer A mixture and vortexed for 10 s. To separate the cytosolic fraction from the nuclei, the sample was centrifuged at 500xg for 5 min at 4°C to pellet the nuclei. The cytosolic fraction was collected into a separate microcentrifuge tube. The pelleted nuclei were then resuspended in Buffer C (20mM HEPES, pH 7.9, 25% glycerol, 1.5mM MgCl, 0.2mM EDTA, 0.1mM Protease Inhibitor, Phosphatase Inhibitor, 0.5mM DTT, and 420mM NaCl) and then vortexed for 30 s. To extract the nuclear proteins (i.e., the nuclear extract), the nuclei were incubated in Buffer C for 1 h while mixing at 4°C. To separate the nuclear extract from the nuclear debris, the mixture was centrifuged at 21,000xg for 20 min at 4°C. The nuclear extract was collected in a separate microcentrifuge tube and flash frozen using liquid nitrogen. Nuclear extracts were stored at −80°C.

Microarray DNA double stranding and PBM protocols are as previously described.[19,40,41] Any changes to the previously published protocols are detailed. Double-stranded microarrays were pre-wetted in HBS (20mM HEPES, 150mM NaCl) containing 0.01% Triton X-100 for 5 min and then de-wetted in an HBS bath. Next the array was incubated with nuclear extract (215$\mu$g of nuclear extract for 4x60K array design or 540$\mu$g of nuclear extract for 4x180K array design) for 1 h in the dark in a binding reaction buffer (20mM HEPES, pH 7.9, 100mM NaCl, 1mM DTT, 0.2mg/mL BSA, 0.02% Triton X-100, 0.4mg/mL salmon testes DNA (Sigma-Aldrich, Catalogue #D7656)). The array was then rinsed in an HBS bath containing 0.1% Tween-20 and subsequently de-wetted in an HBS bath. After the protein incubation, the array was incubated for 20 min in the dark with 20$\mu$g/mL primary antibody for the TF or COF of interest (Table S6). The primary antibody was diluted in 2% milk in HBS. After the primary antibody incubation, the array was first rinsed in an HBS bath containing 0.1% Tween-20 and then de-wetted in an HBS bath. Microarrays were then incubated with 10$\mu$g/mL of either alexa488 or alexa647 conjugated secondary antibody (see Table S6) for 20 min in the dark. The secondary antibody was diluted in 2% milk in HBS. Excess antibody was removed by washing the array twice for 3 min in 0.05% Tween-20 in HBS and once for 2 min in HBS in Coplin jars as described above. After the washes, the array was de-wetted in an HBS bath. Microarrays were scanned with a

GenePix 4400A scanner and fluorescence was quantified using GenePix Pro 7.2. Exported fluorescence data were normalized with MicroArray LINEar Regression.[40]

### CASCADE microarray designs and analyses

A known LPS-responsive segment of the *CXCL10* promoter (hg38; chr4:76023583-76023748) was used in our array design.[23] The genomic region was tiled using overlapping 26-base "target" probe sequences with a 5-base step between sequential tiles. 29 tile probes were needed to span the LPS-responsive *CXCL10* promoter segment. "Target" sequences corresponding to the genomic locus were obtained from the hg38 genome fasta file from Illumina iGenomes using the "fastaFromBed" function from bedtools v2.26.0.[59] For each tile probe and each position along the corresponding 26-base target region, a probe was included in the array design consisting of each possible nucleotide variant (at that position) in order to employ the variant probe analysis approach (see below). A total of 2,291 targets were used to model the *CXCL10* promoter segment (29 tiles + 29 × 3 variant probes x 26 positions). 500 additional 26-base target regions were randomly selected from the hg38 using the bedtools "shuffleBed" function and included in the array design to build a background distribution. Each 26-base target region in the array design was embedded in a larger 60-base PBM probe as follows:

"GCCTAG" 5′ flank – 26-base target region – "CTAG" 3′ flank – "GTCTTGATTCGCTTGACGCTGCTG" double-stranding primer

Each target region was included in its reference (+) orientation as well as the reverse complement (-) orientation. 5 replicate spots of each probe (in each orientation) were included in the final array design. PBM microarray probes, relevant annotation for each, and the experimental results are provided (Table S1). The microarrays were purchased from Agilent Technologies Inc. (AMAID: 085605, format: 8x60K).

To design the screen for differential COF recruitment at SNPs, the lead SNPs uncovered in previous studies were included in our high-throughput screen as follows: 1,446 basal eQTLs[60] (randomly selected from the "classical monocytes" category), 81 caQTL-eQTLs, 11 GWAS caQTLs, 14 GWAS eQTLs, and 160 response eQTLs.[3] Chromosomal coordinates (hg38) for each SNP were obtained using the biomaRt R package from Ensembl.[61] 26-base DNA probe target regions centered at the SNP position (relative to + strand: 13 bases + SNP location + 12 bases) were obtained for each reference (REF) allele using bedtools as above. For each REF allele probe, a probe with the corresponding non-reference (non-REF) allele was also included in the design such that each rsID is represented by a pair of REF and non-REF probes. 500 background target regions were also included using the same procedure as above. The 26-base target regions were embedded in larger 60-base PBM DNA probes as above. 5 replicates of each probe (in both orientations) were included in the final design. The microarrays were purchased from Agilent Technologies Inc. (AMAID: 085920, format: 8x60K).

Each REF/non-REF pair was screened for differential recruitment of p300, SMARCA4, TBL1XR1, RBBP5, and GCN5 as well as differential binding of representative ETS factor PU.1. NextPBM experimental results were preprocessed as above. Z-scores were obtained for each probe as previously described[62] against the distribution of fluorescence intensities obtained at the set of background probes for a given experiment. For each REF and non-REF allele pair in the design, a t test was used to compare the fluorescence intensity distributions between the 5 REF probes and 5 non-REF probes for a given COF/TF assayed. To mitigate the influence of probe orientation-specific effects, t tests were performed independently for each probe orientation with the p values combined using Fisher's method. The Benjamini-Hochberg method was used to adjust the individual p values for a SNP pair (REF and non-REF probes) for multiple hypothesis testing. The fluorescence intensity z-score difference for a given SNP pair (REF and non-REF probes) (termed Δz-score) was computed by subtracting the mean REF z-score from the mean non-REF z-score such that a positive Δz-score represents a gain-of-recruitment introduced by the non-REF allele and a negative Δz-score represents a loss. Scatterplots based on the screening results (Figures 3B and 3C) were plotted using the ggplot2,[58] RColorBrewer,[49] and cowplot[50] R packages. A full data file including the statistics from the high-throughput differential recruitment screen is included in the supplementary materials (Table S2).

REF and non-REF allele pairs exhibiting reproducible significant differential COF recruitment and/or TF binding were selected for this CASCADE array design in order to infer regulators responsible for the differential activity observed. Inclusion criteria was as follows: the difference in recruitment (or binding) of a given COF (or TF) between corresponding REF and non-REF allele probes must have obtained an adjusted p value (q-value) < 0.05 independently in both technical replicates with a concordant direction of effect. Single variant probes for the 26-base target regions (centered at the SNP position – as described above) were generated using the same procedure as above but without the tiling needed to span larger genomic loci such as the *CXCL10* promoter segment used previously. In addition, only 291 background probes were included due to probe number limitations. PBM microarray probes, relevant annotation for each probe, and the experimental results are provided (Table S5). The microarrays were purchased from Agilent Technologies Inc. (AMAID: 086248, format: 4x180K).

Motif modeling using single variant (SV) probes was performed as previously described[19,62,63] for the SNP-QTL sites profiled in detail using CASCADE. For the multi-tile design used to model extended loci such as the LPS-responsive *CXCL10* promoter segment, a weighted mean approach was applied as follows to overlapping positions in order to integrate results across sequential tiles: all variant probes corresponding to a given nucleotide at a given position within the promoter segment were averaged using each probe's corresponding seed (reference genomic) z-score as a weight. Further, if a given SV probe's z-score was above 1.645 (above approximately 95% of the fluorescence intensities obtained using background probe distribution - assuming a normal distribution) and the SV probe's corresponding reference probe z-score was less than or equal to 1.645, the SV probe's z-score was

**CellPress**

**Cell Genomics**
Article

reset to the reference seed value. This procedure ensured that the SV probe modeling approach was used to characterize true genomic recruitment sites and reduce the influence of COF recruitment sites gained specifically via a non-REF variant. Sequence logo plots for the COF recruitment and TF binding motifs were generated using the ggseqlogo R package[51] and arranged using cowplot. The Δz-scores of each nucleotide represent the difference relative to the median z-score obtained across all possible nucleotides at that position and were computed after the weighted averaging procedure described previously. The Δz-score axis limits for the logo tracks (Figures 2 and S2) were determined using the minimum and maximum Δz-scores obtained for a given COF/TF (across experiments within an array design) to enable comparisons across stimulus conditions assuming matched total protein concentrations across experiments.

### CASCADE microarray designs and analyses for TFBS

PBM experiments were performed as previously described (see STAR Methods, CASCADE PBM experimental methods). Primary and secondary antibodies used in this experiment are listed in Table S6. A set of 346 non-redundant TF binding models from the JASPAR 2018 core vertebrate set were obtained using the JASPAR 2018 R bioconductor package. For each transcription factor binding site (TFBS) on the array, a random non-repeating 2 base pad was added to both ends of each consensus sequence to account for possible nucleotide determinants beyond the positions covered by the TFBS. To compensate for size differences between probes, a 34-bp backbone sequence was generated such that the nucleotide at each position was generated randomly with the constraint that sequential positions contain non-repeating nucleotides. Each consensus and SNV sequence generated was then inserted into the backbone sequence beginning at the 5′ end so that the binding site being profiled is located at the end furthest away from the glass side to which the probe is fixed. These 34-base targets were then embedded within a larger probe that is 60 bases long as follows:

GC cap + 34 base target region (TF site or SNV within backbone) + 24 base primer

Multiple background target DNA probes were also included in the design in order to be able to estimate background fluorescence intensities in the experiments. These regions were selected at 34-base genomic segments from the human genome (hg38). For COF recruitment motifs obtained at each TFBS, a minimal seed z-score of 1.5 was enforced. Recruitment energy matrices obtained from profiling COF recruitment of the array were converted to a probability-based matrix using the Botlzmann distribution as described below to be directly compared to published TF binding models (see STAR Methods, Motif similarity analysis).

### Motif similarity analysis

For CASCADE recruitment motifs obtained at the *CXCL10* locus, to simplify the analysis and reduce the number of comparisons, the promoter segment was first separated into 3 motifs broadly corresponding to each previously characterized TF site (ISRE, NF-κB-2, and NF-κB-1). For CASCADE profiling of the SNP-QTLs, a minimal seed z-score of 1.5 was enforced for motif analysis. Recruitment energy matrices obtained from CASCADE cofactor profiling (fluorescence intensity z-scores) were converted to a probability-based matrix using the Boltzmann distribution as previously described[61] to be more directly comparable to previous TF binding models:

$$P_{ik} = \frac{e^{\beta z_{ik}}}{\sum\limits_{k=1}^{4} e^{\beta z_{ik}}}$$

$z_{ik}$ is the z-score for nucleotide variant $k$ at position $i$ within the motif window. $\beta$ transformation parameters for the Boltzmann equation were scaled using the maximum z-score obtained in a given experiment using the following equation in order to account for differences in antibody efficiencies across cofactors:

$$\beta = \frac{30}{\max(z)}$$

Resulting position-weight matrices were compared against the complete HOCOMOCOv11 database[64] of transcription factor binding models (771 total) using TOMTOM from the MEME suite[52] version 5.0.3. Euclidean distance was used as the similarity metric with a relaxed minimal reporting q-value of 0.25 (-dist ed -thresh 0.25). COF recruitment motifs not matching a known motif. For the COF recruitment position-weight matrices included in the motif similarity analysis that did not match a TF binding model at a q-value of 0.25, the experimentally-derived consensus recruitment site was scanned using CisBP[65] and the top-scoring match was reported. These results were flagged as low-stringency matches.

### Identification and processing of THP-1 ChIP-seq data

436 THP-1 ChIP-seq datasets were obtained from the Gene Expression Omnibus (GEO)[66] using custom scripts. The annotations for every dataset (assay type, cell line, assayed molecule) were manually checked to ensure accuracy. The Sequence Read Archive (SRA) files obtained from GEO were analyzed using an automated pipeline. Briefly, the pipeline first runs QC on the FastQ files containing the sequencing reads using FastQC (v0.11.8).[53] If FastQC detects adaptor sequences, the pipeline runs the FastQ files through Trim Galore (v0.4.2),[54] a wrapper script that runs cutadapt (v1.9.1)[55] remove the detected adaptor sequence from the reads. The resulting reads are then aligned to the reference human genome (hg19/GRCh37) using bowtie2 (v2.3.4.1).[56] The aligned reads

(in .BAM format) are then sorted using samtools (v1.8.0)[52] and duplicate reads are removed using picard (v1.89).[67] Finally, peaks are called using MACS2 (v2.1.2).[67] Four different parameter settings are used, in order to capture differences between dataset attributes and qualities: g hs -q 0.01; -g hs -q 0.01 –broad; -g hs -q 0.01–broad–nomodel–extsize 500; -g hs -q 0.01–broad–nomodel–extsize 1000. ENCODE blacklist regions[68] were removed from the peak sets using the hg19-blacklist.v2.bed.gz file available at https://github.com/Boyle-Lab/Blacklist/blob/master/lists/hg19-blacklist.v2.bed.gz. For each dataset, a final set of peaks is established by taking the union of the four peaks sets from the four parameter settings. 33 datasets failed at the download, alignment, or peak calling steps, yielding a total of 403 ChIP-seq peak sets in .BED format.

### Allele-dependent ChIP-seq data analysis

We genotyped the THP-1 monocyte cell line on Illumina OMNI-5 arrays, as previously described.[69] Genotypes were called using the Gentrain2 algorithm within Illumina Genome Studio. Quality control was performed as previously described.[69] Quality control data cleaning was performed in the context of a larger batch of non-disease controls to allow for the assessment of data quality. Briefly, all cell lines had call rates > 99%; only common variants (minor allele frequency > 0.01) were included; and all variants were previously shown to be in Hardy-Weinberg equilibrium in control populations at p > 0.0001.[69] We performed genome-wide imputation using overlapping 150 kb sections of the genome with IMPUTE2[36] and a composite imputation reference panel of pre-phased integrated haplotypes from 1000 Genomes (June 2014). Included imputed genotypes met or exceeded a probability threshold of 0.9, an information measure of 0.5, and the same quality-control criteria described above for the genotyped markers. Regions of the genome with abnormal chromosome counts (i.e., regions that do not have two chromosomes) were removed from consideration using the cnvPartition software package (Illumina Genome Studio) with default parameter settings, due to potential confounding effects on allelic read imbalance analysis. 389 of the ChIP-seq datasets had at least one allelic result. Of these, 297 had at least one allelic result for a genetic variant examined on the PBM arrays.

Cell line annotations were double checked by examining read counts at variants identified as heterozygous by the genotyping arrays. Specifically, for each dataset, we examined all heterozygotes with at least five sequencing reads. We then calculated the fraction of these variants with exactly zero reads on the weak allele (i.e., the allele with fewer mapped sequencing reads). Zero weak allele reads is a hallmark of a mis-annotated cell line, since a variant that is thought to be a heterozygote, but in reality is a homozygote, will always exhibit zero weak allele reads. Any dataset with 45% or more of its heterozygous variants displaying at least 5 strong allele reads and exactly zero weak allele reads was flagged as a likely mis-annotation by the original dataset producers and removed from downstream analyses (59 of the 297 datasets were removed). This cutoff was chosen based on comparisons between purposely matched and purposely mis-matched genotyping array/ChIP-seq experiment pairs. As a result, 238 datasets were used for allelic analyses for the THP-1 cell line.

To identify possible mechanisms underlying the allelic behavior observed in the PBM experiments, we applied our MARIO[36] method to the THP-1 ChIP-seq dataset collection described above. In brief, MARIO identifies common genetic variants that are (1) heterozygous in the assayed cell line and (2) located within a peak in a given ChIP-seq dataset. It then examines the sequencing reads that map to each heterozygote in each peak for imbalance between the two alleles. Results with MARIO Allelic Reproducibility Score (ARS) values > 0.4 were considered allelic, following our previous study.[36] For comparison against the differential COF recruitment screening data, SNPs with overlapping significant COF recruitment differences and evidence of allelic imbalance in the ChIP-seq datasets were identified using a custom R script and the different SNP set intersection sizes were plotted using the UpSetR package.[57] The following features previously investigated using ChIP-seq in THP-1 cells were considered general chromatin features for these comparisons: H3K27ac, PAF1, POLR2A, CTCF, H3K4me3, POLR3D, KMT2B, RAD21, H3K4me2, LEO1, NR3C1, RCOR1, and H3K27me3.

### ENCODE ChIP-seq data analysis

ChIP-seq datasets were downloaded from the ENCODE consortium site (https://www.encodeproject.org/). To analyze binding of our COFs in the same cell type, Bed data files were downloaded for TBL1XR1 (ENCFF868SWL), RBBP5 (ENCFF666PCE), EP300 (ENCFF755HCK), and SMARCA4 (ENCFF267OGF), for binding data in K562 cells. ChIP-seq regions were considered overlapping if a single nucleotide of two regions overlapped. The UpSet plot to illustrate the overlaps of different regions was generated using the R software "upset" function.

## QUANTIFICATION AND STATISTICAL ANALYSIS

### CASCADE analyses for genomic loci

For each REF and non-REF allele pair included in the SNP screen, a t test was used to compare the fluorescence intensity distributions between 5 REF and non-REF probes for a given COF/TF assayed. To mitigate the influence of probe orientation-specific effects, t tests were performed independently for each probe orientation with the p values combined using Fisher's method. The Benjamini-Hochberg method was used to adjust the individual p values for a SNP pair (REF and non-REF probes) for multiple hypothesis testing. The fluorescence intensity z-score difference for a given SNP pair (REF and non-REF probes) (termed Δz-score) was computed by subtracting the mean REF z-score from the mean non-REF z-score such that a positive Δz-score represents a gain-of-recruitment introduced by the non-REF allele and a negative Δz-score represents a loss (Figures 3, 4, and 5).

**CellPress**

**Cell Genomics**
Article

### Determination of SNPs to include for CASCADE

REF and non-REF allele pairs exhibiting reproducible significant differential COF recruitment and/or TF binding were selected for this CASCADE array design in order to infer regulators responsible for the differential activity observed. Inclusion criteria was as follows: the difference in recruitment (or binding) of a given COF (or TF) between corresponding REF and non-REF allele probes must have obtained an adjusted p value (q-value) < 0.05 independently in both technical replicates with a concordant direction of effect (Figure 3).

### Motif similarity analysis

To determine the identity of the underlying motif, recruitment energy matrices were first converted to a probability-based matrix (PWM) using the Boltzmann distribution as previously described.[61] The PWMs were compared against the HOCOMOCOv11 database58 of transcription factor binding models (771 total) using TOMTOM from the MEME suite59 version 5.0.3. Euclidean distance was used as the similarity metric with a relaxed minimal reporting q-value of 0.25 (-dist ed -thresh 0.25). COF recruitment motifs not matching a known motif. For the COF recruitment position-weight matrices included in the motif similarity analysis that did not match a TF binding model at a q-value of 0.25, the experimentally-derived consensus recruitment site was scanned using CisBP60 and the top-scoring match was reported. These results were flagged as low-stringency matches (Figures 2, 3, 4, and 5).

