## [Document S3. Transparent peer review records for Bray et al. · Cell Genomics]

CASCADE: high-throughput characterization of regulatory complex binding altered by non-coding variants

David Bray^{1,2,3}, Heather Hook^{1,2}, Rose Zhao^{1,2}, Jessica L. Keenan^{1,2,3}, Ashley Penvose^{1,2}, Yemi Osayame^{1,2}, Nima Mohaghegh^{1,2}, Xiaoting Chen⁴, Sreeja Parameswaran⁴, Leah C. Kottyan^{4,7,8}, Matthew T. Weirauch^{4,5,6,7}, Trevor Siggers^{1,2,*}

Summary

Initial submission: Received : August 25th 2020

Scientific editor: Orli Bahcall

First round of review: Number of reviewers: 2
Revision invited : February 21st 2021
Revision received : August 20th 2021

Second round of review: Number of reviewers: 2
Accepted : January 19th 2022

Data freely available:

Code freely available:

This transparent peer review record is not systematically proofread, type-set, or edited. Special characters, formatting, and equations may fail to render properly. Standard procedural text within the editor's letters has been deleted for the sake of brevity, but all official correspondence specific to the manuscript has been preserved.

Referee reports, first round of review

Reviewer #1

Overview

This article introduces and evaluates CASCADE, an experimental method for determining the effects of non-coding DNA polymorphisms on the recruitment of gene regulatory complexes such as p300. The method is a variant on protein binding microarrays in which nuclear lysate is hybridized to an array of DNA probes. The array is then incubated with an antibody to any protein of interest to determine which probes it is associated with. One application focuses on a specific cis-regulatory element (CRE) and uses probes that tile the CRE along with all probes that differ from the tiling probes at one position. In this application, analysis of the signals from the probes yields hypothesized binding motifs and binding locations within the target CRE. Searching a database of known motifs may yield matches to one or more DNA-binding TFs that are hypothesized to recruit the target protein to the DNA. Another application screens probes designed to match each allele at suspected expression quantitative-trait loci (eQTLs). Probes yielding significant effects can then be followed up with additional variants of those probes on separate arrays to yield hypothesized binding motifs and TFs that may mediate the binding.

The major claims are:

1. The method works – it has sufficient sensitivity and specificity to be useful.
2. The method is high throughput.
 - a. “The CASCADE approach addresses the current need for high-throughput methods to determine the impact of NCVs on TF-COF regulatory complex binding.”
 - b. “CASCADE provides the first high-throughput approach to both discover and characterize TF-COF complexes impacted by NCVs”
3. The method can be used to identify the TF that links the SNP locus to the regulatory cofactor by inferring a PWM binding model and comparing it to databases of known TF-PWM associations.

Originality

CASCADE is distinct from existing methods such as classic protein-binding microarrays (PBMs), in which a purified DNA-binding TF is hybridized to a DNA array. It is also distinct from ChIP-Seq against TFs or cofactor complexes because it uses synthesized DNA designed to answer specific questions, rather than relying on the native genomes of cells. The authors should clarify the relationship of this work to their 2019 paper “NextPBM: a platform to study cell-specific transcription factor binding and cooperativity” in *Nucleic Acids Research*, which they cite in the Methods but not in the Introduction or Discussion.

Importance to researchers in its field

In support of its importance, the paper states that “by combining the flexibility of the fully customizable, array-based approach with the use of nuclear extracts, CASCADE can be used to analyze tailored lists of target NCVs in a cell state-specific manner.” I agree that this fills a niche that is left between methods such as ChIP-Seq, which is limited by the diversity present in the target cells, and PBMs hybridized with purified proteins rather than nuclear extracts. By extracting nuclei from cells of a particular type or in a particular condition, users of this method can focus on the most relevant cellular context.

The paper also states that “...in CASCADE, TF-COF complexes are profiled, as opposed to TFs alone, allowing us to link NCVs with the diverse biological functions mediated by COFs, such as histone modifications and chromatin remodeling. This provides an additional level of biological annotation beyond current approaches that focus on TF binding alone.” I agree that this additional information

about the mechanisms of regulation, not available from ChIP against DNA-binding TFs, would be somewhat useful.

A strength and a limitation of the method is that it requires targeting a particular regulatory complex. This is a strength compared to ChIP-Seq against TFs and traditional PBMs because the same complex can work with many different TFs. In principle, one can test the significance of a SNP more comprehensively using fewer antibodies targeting fewer proteins. This is also a limitation because only SNPs that function via the targeted complexes will be detected as functional. It would be helpful if the authors could discuss how many such complexes are commonly used in transcriptional regulation and how many TFs work by transcriptional regulatory domains intrinsic to their peptide sequence rather than common, non-covalently associated cofactors. To assess the utility of this method accurately, users need to know how many experiments with how many different antibodies are required to achieve a certain level of coverage of TF space. The results show that ~10-15% of eQTLs could be validated by this method. This may indicate that many of the non-validated eQTLs work through intrinsic regulatory domains or through cofactors that were not tested.

If its claims to efficacy are substantiated, this paper would be a solid contribution to regulatory systems genomics and the emerging field of experimental follow-up to candidate SNPs identified by GWA studies. It would provide an additional, complementary tool in the diverse toolkits of these fields.

Importance to researchers outside the field

If its claims are substantiated, this paper would be of some interest to anyone working in the rather large field of human genetics.

Rigorous methodology and substantial evidence for its conclusions

I have no concerns about the rigor of the experimental results.

Claim 1: The method has sufficient sensitivity and specificity to be useful. The evidence in support of this claim is based on a detailed dissection of the wellstudied promoter of CXCL10 and on a survey of ~1600 eQTLs. Although I am not familiar with the literature on CXCL10, I find it plausible that the CASCADE experiment identified the important features of this promoter, including the sites to which p300 and RBBP5 are recruited, the binding motifs of the TFs that recruit them, and the identities of those TFs. As mentioned above, the eQTL experiment validated 10-15% of the ~1,600 eQTLs tested as differentially recruiting one of 5 cofactors. It is difficult to know how sensitive or specific this result is. I would assume the false discovery rate of the initial eQTL detection is lower than 85%, so presumably there are truly functional eQTLs that were not detected in the CASCADE experiment. This could be because the experimental method was not sufficiently sensitive to detect them or because they work via cofactors that were not tested, via TFs with intrinsic regulation domains, or in cell types or conditions not tested.

Claim 2: The method is high throughput. High throughput is relative. Compared to studying one TF at a time, the method is more efficient, because many TFs recruit the same cofactors. However, one of the two experiments reported studied a single promoter that is ~200 bp long. It is hard to argue that such an application is high throughput, in the sense of being able to efficiently screen the many SNPs in the many loci typically linked to a phenotype in human genetics studies. The other demonstration tested 1,600 suspected eQTL for differential recruitment of 5 cofactors. 1,600 is plausibly on the scale of variants found within a single GWAS peak. However, these 1,600 are not tested comprehensively. It's

likely that many of them are eQTLs that work by mechanisms other than the 5 tested cofactors. The claim has some merit, but this method is not high throughput in the sense of being easily applied genome wide.

Claim 3: The method can be used to identify the TF that links the SNP to the regulatory cofactor. The demonstration using the well-studied CXCL10 promoter does not provide strong evidence for this claim. Not only is the promoter well studied, the TFs involved are very well studied. It is not clear how this will generalize to identifying one or even a handful likely TFs out of the 1500 TFs encoded in the human genome.

Summary

CASCADE seems to fill a niche that is not already densely populated by other methods. It will be a useful addition to the functional genomics toolkit. It is not clear that it can be applied in genome-wide discovery studies, but it would likely be useful for studying potential causal SNPs in a locus linked to a phenotype.

Other comments

“TFs function by recruiting COFs to DNA, which subsequently alter gene expression through diverse mechanisms.” This seems to say that TFs with their own, intrinsic activation or repression domains are negligible, which is not a mainstream view and requires justification.

Reviewer #2

Review of Bray et al.

In this manuscript, the authors describe a novel and innovative method that can measure binding of transcriptional co-factors on a large number of dsDNA templates. The strength of the approach is the large number of sequences analyzed, allowing the authors to study a large set of sequence variants. The main weakness is the use of free DNA templates instead of nucleosomal DNA, limiting the approach somewhat. I find this acceptable for this well-written paper that describes an important technology.

The manuscript is well written and clear. I recommend publication after the following issues are addressed:

1) The mapping of motifs to transcription factors is too simplistic. It is generally not possible to tell the exact transcription factor that binds to DNA based on a motif. The authors should refrain from making overly specific predictions, as there is always a motif that is closest to another motif, but in many cases the differences are not significant (the precise assignment is then based on noise and not signal). This part would also benefit from analysis of expression of the TF proteins, and a more balanced discussion.

2) Many co-factors are recruited by the same transcription factors in the assay. The authors should clarify why this is the case. Are the co-factors recruited directly but relatively non-specifically by the transcription factors, or is their recruitment to DNA associated with binding of large protein complexes such as mediator? Finally, it is possible that the recruitment resembles a process that happens in cells, where transcription factors block binding of other proteins such as histones or DNA damage response proteins from the nuclear extract that would mask the DNA and prevent co-factor association. The cell-derived proteins that bind to the arrays can easily be determined by mass-spectrometry, and the

accessibility of the array can also be assessed using nucleases, transposases, dsDNA binding domains or specific transcription factors.

3) Is the relatively non-specific recruitment and the more specific associations also observed in vivo? As the sequences used are derived from the genome, the results can easily be compared to ChIP-seq and other methods to see whether the co-factors are commonly co-localized in cells, and how well the in vitro approach predicts in vivo binding.

4) Stylistically, the manuscript mainly focuses on technology, and clearly demonstrates that the methodology yields results that are consistent with existing knowledge. It does not, however, contain a major new biological discovery. This is acceptable, but in addition to the strengths of the method, its limitations should also be clearly stated. First, the authors should describe more clearly how much sequence space can be realistically covered by the arrays. Second, they should discuss the limitations in terms of how long sequences can be studied. Third, more discussion about biological relevance is needed. Analysis of promoter sequences is clearly feasible using the method, as promoters are commonly located in accessible chromatin regions. However, most regulatory and disease-linked variants are located far from transcription start sites. Therefore, the authors should clearly state in the discussion that the methodology will lead to many binding effects that may not be biologically important as the sequences studied are located in closed chromatin in cells, and provide the reader a way to understand how the method could be combined with other genomics methods for analysis of all non-coding variants.

Authors' response to the first round of review

Response to Reviewer Comments:

Reviewer 1: This article introduces and evaluates CASCADE, an experimental method for determining the effects of non-coding DNA polymorphisms on the recruitment of gene regulatory complexes such as p300. The method is a variant on protein binding microarrays in which nuclear lysate is hybridized to an array of DNA probes. The array is then incubated with an antibody to any protein of interest to determine which probes it is associated with. One application focuses on a specific cis-regulatory element (CRE) and uses probes that tile the CRE along with all probes that differ from the tiling probes at one position. In this application, analysis of the signals from the probes yields hypothesized binding motifs and binding locations within the target CRE. Searching a database of known motifs may yield matches to one or more DNA-binding TFs that are hypothesized to recruit the target protein to the DNA. Another application screens probes designed to match each allele at suspected expression quantitative-trait loci (eQTLs). Probes yielding significant effects can then be followed up with additional variants of those probes on separate arrays to yield hypothesized binding motifs and TFs that may mediate the binding.

The major claims are:

1. The method works – it has sufficient sensitivity and specificity to be useful.
2. The method is high throughput.
 - a. “The CASCADE approach addresses the current need for high-throughput methods to determine the impact of NCVs on TF-COF regulatory complex binding.”

b. b. "CASCADE provides the first high-throughput approach to both discover and characterize TF-COF complexes impacted by NCVs"

3. The method can be used to identify the TF that links the SNP locus to the regulatory cofactor by inferring a PWM binding model and comparing it to databases of known TF-PWM associations.

Reviewer 1: Originality: CASCADE is distinct from existing methods such as classic protein binding microarrays (PBMs), in which a purified DNA-binding TF is hybridized to a DNA array. It is also distinct from ChIP-Seq against TFs or cofactor complexes because it uses synthesized DNA designed to answer specific questions, rather than relying on the native genomes of cells. The authors should clarify the relationship of this work to their 2019 paper "NextPBM: a platform to study cell-specific transcription factor binding and cooperativity" in *Nucleic Acids Research*, which they cite in the Methods but not in the Introduction or Discussion.

Author Response: We agree that there needs to be further clarification about the relationship between this work and our 2019 paper "NextPBM: a platform to study cell-specific transcription factor binding and cooperativity" in *Nucleic Acids Research*. To address this, we have added the following text to clarify this in the introduction section when we introduce the CASCADE method:

"This approach builds upon our recent work using nuclear extracts on PBMs to study TFDNA binding in a more endogenous, cell-specific context [PMCID: PMC6451091]; however, rather than profiling TF binding, in CASCADE we profile recruitment of COFs to DNA by underlying TFs."

Reviewer 1: Importance to researchers in its field: In support of its importance, the paper states that "by combining the flexibility of the fully customizable, array-based approach with the use of nuclear extracts, CASCADE can be used to analyze tailored lists of target NCVs in a cell state-specific manner." I agree that this fills a niche that is left between methods such as ChIPSeq, which is limited by the diversity present in the target cells, and PBMs hybridized with purified proteins rather than nuclear extracts. By extracting nuclei from cells of a particular type or in a particular condition, users of this method can focus on the most relevant cellular context. The paper also states that "...in CASCADE, TF-COF complexes are profiled, as opposed to TFs alone, allowing us to link NCVs with the diverse biological functions mediated by COFs, such as histone modifications and chromatin remodeling. This provides an additional level of biological annotation beyond current approaches that focus on TF binding alone." I agree that this additional information about the mechanisms of regulation, not available from ChIP against DNA-binding TFs, would be somewhat useful. A strength and a limitation of the method is that it requires targeting a particular regulatory complex. This is a strength compared to ChIP-Seq against TFs and traditional PBMs because the same complex can work with many different TFs. In principle, one can test the significance of a SNP more comprehensively using fewer antibodies targeting fewer proteins. This is also a limitation because only SNPs that function via the targeted complexes will be detected as functional. It would be helpful if the authors could discuss how many such complexes are commonly used in transcriptional regulation and how many TFs work by transcriptional regulatory domains intrinsic to their peptide sequence rather than common, non-covalently associated cofactors. To assess the utility of this method accurately, users need to know how many experiments with how many different antibodies are required to achieve a certain level

of coverage of TF space.

Author Response: We agree with the Reviewer that determining the number of TFs that function via COF interactions will be helpful in determining the *coverage* of our approach, and how many TFs we are likely to be able to identify. Unfortunately, determining how most human TFs function remains an outstanding problem, which is exacerbated by the fact that TF-COF interactions are cell-state specific and interactions with COFs are considerably under sampled. However, we believe that one point of confusion might be the interpretation of what we mean by COF. By COF we mean any recruited transcriptional regulator that interacts with TFs, including Mediator and other general TFs. We have attempted to make this clearer now in the main text: “One way TFs function is by recruiting COFs to DNA, which subsequently alter gene expression through diverse mechanisms including interactions with the transcriptional machinery (e.g., Mediator), histone modification (e.g. EP300 histone acetylase), DNA modification (e.g., DNA methylases), chromatin remodeling (e.g., SWI/SNF-type complexes), or bridging to other COFs (e.g., BRD4) (Figure 1a)18.”

Interpreting COFs more broadly, we believe that many TFs function at least in part through such COF interactions. From Staller et al. (PMID: 29525204) “...activation domains are poorly conserved, intrinsically disordered, and bind structurally diverse coactivators”. Finally, to address this issue of coverage we examined the TF-COF interactions present in two large public PPI databases (APID and BioGRID). We found that on average there were 10 COF interactions per TF, but with rather large tails in which some well-studied TF families had more, p53 (~120), Rel/NF- κ B family (~40), and Nuclear Receptors (~35). We have added text in the Discussion related the concept of coverage:

“By performing our screen with a panel of broadly acting COFs from diverse functional categories, we can profile a large fraction of potentially functionally relevant TF-COF complexes. Our analyses of the *CXCL10* promoter (Figure 2), SNP loci (Figure 4), and expanded set of synthetic loci (Supplementary Figure 6) indicate that a range of different TF families are being identified, despite using only a small number of COFs. However, it remains unclear how many of the ~1600 human TFs we can profile using this approach, but future studies using expanded sets of COFs will help to clarify this point and establish whether some TFs are more efficiently profiled via COFs than others.”

Reviewer 1: Importance to researchers outside the field: If its claims are substantiated, this paper would be of some interest to anyone working in the rather large field of human genetics.

Rigorous methodology and substantial evidence for its conclusions: I have no concerns about the rigor of the experimental results.

Author Response: We thank the reviewer and believe that it will be of interest to researchers in the field of human genetics.

Reviewer 1: Claim 1: The method has sufficient sensitivity and specificity to be useful. The evidence in support of this claim is based on a detailed dissection of the well-studied promoter of *CXCL10* and on a survey of ~1600 eQTLs. Although I am not familiar with the literature on *CXCL10*, I find it plausible that the CASCADE experiment identified the important features of this promoter, including the sites to which p300 and RBBP5 are recruited, the binding motifs of the TFs that recruit them, and the identities of those TFs. As mentioned above, the eQTL experiment validated 10-15% of the ~1,600 eQTLs tested as differentially

recruiting one of 5 cofactors. It is difficult to know how sensitive or specific this result is. I would assume the false discovery rate of the initial eQTL detection is lower than 85%, so presumably there are truly functional eQTLs that were not detected in the CASCADE experiment. This could be because the experimental method was not sufficiently sensitive to detect them or because they work via cofactors that were not tested, via TFs with intrinsic regulation domains, or in cell types or conditions not tested.

Author Response: We agree with the Reviewer and believe that many eQTLs were not identified in our assay because they may function via COFs that were not profiled, or that they function in cell types/conditions not tested. We have added text (discussed above) to the discussion about coverage and believe that future studies will help us to identify COFs that can provide the broadest possible set of TF-COF complexes. We also believe that the cell state specificity of the approach is a positive aspect of our approach, enabling one to identify cell specific regulatory complexes, but it does come at a price that one needs to know the correct condition, as is true with any experimental approach (i.e., reporter assays, etc.).

Reviewer 1: Claim 2: The method is high throughput. High throughput is relative. Compared to studying one TF at a time, the method is more efficient, because many TFs recruit the same cofactors. However, one of the two experiments reported studied a single promoter that is ~200 bp long. It is hard to argue that such an application is high throughput, in the sense of being able to efficiently screen the many SNPs in the many loci typically linked to a phenotype in human genetics studies. The other demonstration tested 1,600 suspected eQTL for differential recruitment of 5 cofactors. 1,600 is plausibly on the scale of variants found within a single GWAS peak. However, these 1,600 are not tested comprehensively. It's likely that many of them are eQTLs that work by mechanisms other than the 5 tested cofactors. The claim has some merit, but this method is not high throughput in the sense of being easily applied genome wide.

Author Response: We agree with the Reviewer that the term high throughput is relative and could lead to some confusion. We believe that our approach is high throughput in the sense that we can test the binding of our profiled COFs to ~1700 REF/SNP pairs. Therefore, by profiling 5 COFs to 1700 REF/SNPs we are testing 10,500 interactions in a cell-specific manner. This is on par with early high-throughput reporter assay approaches in terms of numbers of tested elements, or even the traditional PBM assay which was widely characterized as a highthroughput TF-DNA binding assay. We have amended the text in the Abstract to clarify this point now, focusing on the high throughput *testing* as opposed to *annotation*:
“Our customizable CASCADE approach will enable the high-throughput *testing* of the molecular mechanisms of NCVs for diverse biological applications.”
We also believe the term *comprehensive* in our acronym might cause some confusion. Rather, the term describes the fact that we comprehensively test all DNA variants, not that all TF-COF complexes are comprehensively tested. To address this point, we have changed the terminology in our acronym to remove the term ‘comprehensive’. It now reads: CASCADE (Customizable Approach to Survey Complex Assembly at DNA Elements).

Reviewer 1: Claim 3: The method can be used to identify the TF that links the SNP to the regulatory cofactor. The demonstration using the well-studied *CXCL10* promoter does not provide strong evidence for this claim. Not only is the promoter well studied, the TFs involved are very well studied. It is not clear how this will generalize to identifying one or even a handful

likely TFs out of the 1500 TFs encoded in the human genome.

Author Response: We agree with the Reviewer that it remains unclear how many TFs will be profiled using this approach. However, as a technical platform, PBMs have been used to profile hundreds of distinct TFs from diverse families (Uniprobe Database:

http://the_brain.bwh.harvard.edu/uniprobe/); therefore, we do not foresee any technical limitation to many TFs being able to bind the microarray.

A more relevant issue, related to one discussed above, is how many TFs we can identify by profiling recruitment of a particular COF. This relates to the more fundamental question of how many TFs interact with a given COF in a particular cell state, the answer to which is not clear. To this end, in terms of our approach, our focus has been to take a *function-first approach* and start with regulatory COFs known to be important in gene regulation and to identify SNPs that disrupt binding of these COFs. We fully appreciate that we will likely need to profile a larger number of COFs to find more TFs, and future studies will provide clarity on this issue. Here we present the basic platform and demonstrate the feasibility of profiling the binding of TF-COF complexes in this manner, which we think will be of interest to the genetics community. Finally, to address the question of how many TFs *can be profiled* using these COFs in this cell type, we have conducted new experiments using a re-designed array. Briefly, we designed an array with representative consensus sites for ~350 TF families using the core set of human TF motifs in the JASPAR database. For each consensus site we also included all singlenucleotide variants to allow COF recruitment motifs to be generated, as is done in our CASCADE approach. This array design allowed us to look more broadly at TFs, rather than those that would only bind to our CXCL10 loci or the profiled SNPs. We have included an analysis of these results now in the main text (see R2R below, Reviewer 2 point #4). We have also directly commented on this issue in the Discussion (new text described above addressing the point titled “Importance to research in the field”).

Reviewer 1: Summary: CASCADE seems to fill a niche that is not already densely populated by other methods. It will be a useful addition to the functional genomics toolkit. It is not clear that it can be applied in genome-wide discovery studies, but it would likely be useful for studying potential causal SNPs in a locus linked to a phenotype.

Author Response: We thank the Reviewer and agree that the approach will be useful for studying potential causal SNPs.

Reviewer 1: Other comments: “TFs function by recruiting COFs to DNA, which subsequently alter gene expression through diverse mechanisms. This seems to say that TFs with their own, intrinsic activation or repression domains are negligible, which is not a mainstream view and requires justification.”

Author Response: We agree with the Reviewer that TFs intrinsic activation and repression domains are critical to their function. However, how these domains function remains less clear (see above) but is likely due in many cases to interactions with coactivators and corepressors, which are all examples of COFs. We have addressed this point above and amended our description of COF to better reflect this broader definition.

Reviewer 2: In this manuscript, the authors describe a novel and innovative method that can measure binding of transcriptional co-factors on a large number of dsDNA templates. The

strength of the approach is the large number of sequences analyzed, allowing the authors to study a large set of sequence variants. The main weakness is the use of free DNA templates instead of nucleosomal DNA, limiting the approach somewhat. I find this acceptable for this well written paper that describes an important technology. The manuscript is well written and clear. I recommend publication after the following issues are addressed:

1. The mapping of motifs to transcription factors is too simplistic. It is generally not possible to tell the exact transcription factor that binds to DNA based on a motif. The authors should refrain from making overly specific predictions, as there is always a motif that is closest to another motif, but in many cases the differences are not significant (the precise assignment is then based on noise and not signal). This part would also benefit from analysis of expression of the TF proteins, and a more balanced discussion.

Author Response: We agree with the Reviewer on this point. We have altered the text to make clear that our predictions are at the level of TF family, as is done routinely for TF inference in the analysis of genomic datasets. Further, we only indicate specific TFs when we perform follow-up experiments in which the TFs are directly tested (e.g., IRF3, RelA, IRF2, etc). The text added to the manuscript is as follows:

(Text added to the Introduction):

“... by assaying COF recruitment to all single-nucleotide variants of a target DNA sequence we can determine a COF *recruitment* motif. Comparison of the COF recruitment motif against TF-motif databases allows us to annotate the recruiting TF to the level of TF family, as is frequently done in the analysis of genomic datasets, such as CHIP-seq or ATAC-seq data (Figure 1b).”

(Text altered in the Discussion to elaborate on this point as requested):

“...DNA-bound TFs are identified indirectly via motif comparison against COF recruitment motifs. Therefore, our approach is unable to distinguish between related TFs that have similar DNA binding motifs, which is often the case for members of the same TF family. Integrating CASCADE results with other data types, such as gene expression or protein abundance data, can provide a way to refine our TF annotations for follow-up experiments, as was done for the IRF and NF- κ B-family proteins in our *CXCL10* promoter analysis.”

2. Many co-factors are recruited by the same transcription factors in the assay. The authors should clarify why this is the case. Are the co-factors recruited directly but relatively nonspecifically by the transcription factors, or is their recruitment to DNA associated with binding of large protein complexes such as mediator?

Author Response: We were also intrigued by this finding, and now more directly address it in the manuscript. First, to address the specificity of COF recruitment we sought to directly assess the specificity of COF recruitment. To do this we sought to determine whether the recruitment was a result of the particular TFs binding to these specific genomic loci, or whether this was a more general feature of many TFs. To address this, we designed a new microarray in which ~350 consensus TF binding sites (along with exhaustive single-nucleotide variants) were included on the array and then we profiled the recruitment of each COF to this diverse panel of TF binding sites. Second, we examined available CHIP-seq data from the ENCODE consortium in which our COFs were assayed in the same cell types and examined their overlapping profiles. We have now included this analysis in the main text:

“Our analyses also revealed that a number of COFs share similar motif profiles,

suggesting that at these loci they are recruited by a common TF or by related TF-family members. We first examined whether this is a result of the particular QTLs analyzed or a more general feature of these COFs. To examine COF specificity in our CASCADE assay, we designed a new microarray to profile COF recruitment to a panel of ~350 consensus TF sites (and single-nucleotide variants) derived from motifs in the JASPAR database (see Methods). Profiling COF recruitment to this diverse panel of sites revealed common recruitment to some TF families as seen on our QTL loci (e.g., ETS and IRF families), but also revealed COF-specific recruitment preferences (Supplementary Figure 6). For example, only TBL1XR1 was strongly recruited to Nuclear Receptor-family sites, and only RBBP5 was strongly recruited to POU-family sites (Supplementary Figure 6). To determine whether common recruitment preferences might be a result of COFs known to mediate interactions between COFs and TFs we also profiled recruitment of Mediator subunit MED1 and the scaffold protein BRD4; however, we found that these COFs exhibited their own distinct profiles suggesting that Mediator or other scaffolds proteins do not likely explain common recruitment preferences. These results suggest that recruitment to common TFs is not a general feature of our assay but is an inherent feature of these COFs and the loci examined. Next, to examine COF recruitment specificity *in vivo*, we examined available ENCODE ChIP-seq datasets where four of the COFs in our study (EP300, SMARCA4, TBL1XR1, RBBP5) were profiled in K562 cells [PMCID: PMC3439153]. We found that these COFs exhibited highly overlapping binding profiles (Supplementary Figure 7). For example, of the 24,744 EP300 binding loci, 83% overlapped with binding of these other three COFs. Further, 38% of EP300 binding overlapped with both SMARCA4 and TBL1XR1, the two COFs that exhibited similar motif profiles in our analyses. Together, these results suggest that similar COF recruitment profiles seen in our analysis is a result of direct recruitment of these COFs by individual TFs or TF-family members, and that this is consistent with overlapping recruitment profiles seen *in vivo*."

Finally, it is possible that the recruitment resembles a process that happens in cells, where transcription factors block binding of other proteins such as histones or DNA damage response proteins from the nuclear extract that would mask the DNA and prevent co-factor association. The cell-derived proteins that bind to the arrays can easily be determined by mass spectrometry, and the accessibility of the array can also be assessed using nucleases, transposases, dsDNA binding domains or specific transcription factors.

Author Response: We are a little uncertain what the Reviewer is referring to regarding how "proteins from the nuclear extract that would mask the DNA and prevent co-factor association" and to what part of our manuscript this refers. However, we do not believe that masking of DNA by other proteins happens to any appreciable extent. First, when profiling TF binding using nuclear extracts, we have had robust results for a number of different TFs that agree with known DNA binding specificities, suggesting that they are binding the DNA in a predictable manner. Second, in our previous manuscript introducing the nextPBM approach, we specifically examined the ability to assay the binding of two different ETS factors from the same extracts, and we found that they bound DNA without appreciably affecting each other on the array. We believe that this is a consequence of the fact that there is an excess of DNA on each spot compared to the concentrations of TFs present in the nuclear extracts, which means that TFs do not appreciably occlude the binding of each other. For these reasons, we do not believe that steric hindrance/occlusion is strongly affecting our results. Regarding the idea that cell-derived proteins that are bound to the array can be easily identified using MS, we have several concerns. First, relieving the proteins from the array by

protease digestion is difficult and would not likely produce enough material for MS. But perhaps more importantly, it would no longer be specific to any particular sequence, and therefore, it would be hard to interpret any findings. Finally, in terms of development of assays using nucleases and transposases might be possible in principle but would require the development of a whole new assay, and we are not completely clear what we would gain from these experiments given the points described above.

3. Is the relatively non-specific recruitment and the more specific associations also observed *in vivo*? As the sequences used are derived from the genome, the results can easily be compared to ChIP-seq and other methods to see whether the co-factors are commonly co-localized in cells, and how well the *in vitro* approach predicts *in vivo* binding.

Author Response: We examined available ChIP-seq datasets for these factors and have included this analysis in the main text (described above for point #2).

4. Stylistically, the manuscript mainly focuses on technology, and clearly demonstrates that the methodology yields results that are consistent with existing knowledge. It does not, however, contain a major new biological discovery. This is acceptable, but in addition to the strengths of the method, its limitations should also be clearly stated. First, the authors should describe more clearly how much sequence space can be realistically covered by the arrays. Second, they should discuss the limitations in terms of how long sequences can be studied. Third, more discussion about biological relevance is needed. Analysis of promoter sequences is clearly feasible using the method, as promoters are commonly located in accessible chromatin regions. However, most regulatory and disease-linked variants are located far from transcription start sites. Therefore, the authors should clearly state in the discussion that the methodology will lead to many binding effects that may not be biologically important as the sequences studied are located in closed chromatin in cells and provide the reader a way to understand how the method could be combined with other genomics methods for analysis of all non-coding variants.

Author Response: We thank the Reviewer for their careful consideration. We have now expanded our 'limitations' section in the Discussion to address these key points: the sequence coverage and the issue of chromatin. The following is the excerpt from the expanded discussion of the limitations:

"First is the issue of how many loci can be analyzed, either CREs or SNP loci. Using the Agilent 4x180K microarray platform discussed here, we can profile 1,091 bp of CRE sequence in a single experiment at the single-nucleotide level. However, to increase coverage in our SNP screen we described a two-stage approach in which we first screened binding to pairs of REF/non-REF loci (up to 8,467 can be screened) and then assayed a reduced set significant loci at the single-nucleotide level to identify COF motifs (up to 214 loci can be analyzed at this level). A similar, two-stage approach could also be used to more efficiently profile TF-COF binding to CREs to achieve greater coverage. By first profiling binding using 5-bp overlapping tile sequences we could screen 84,691bp of CRE sequence from the genome, and then assay 214 significantly bound loci at the single-nucleotide level. Second is the issue of interpreting the binding results when chromatin is not present on our arrays. As with all *in vitro* assays, our approach is meant to characterize the possible complexes that can assemble on each DNA loci

when DNA is accessible. While these loci may be inaccessible in the native chromatin environment, one can integrate this approach with chromatin maps to bias the loci being analyzed. For example, one might first identify open chromatin regions using ATAC-seq or by profiling histone marks, and then follow up with a CASCADE analysis to profile the TF-COF binding and impact of SNPs.”

Referees' report, second round of review

Reviewer #1

(No further comments)

Reviewer #2

The authors have addressed all of my concerns and added interesting new experiments that show co-factor recruitment specificity. I recommend publication of this important work.

Authors' response to the second round of review

Reviewer #1: (No further comments)

Reviewer #2: The authors have addressed all of my concerns and added interesting new experiments that show co-factor recruitment specificity. I recommend publication of this important work.

Author's Response: We appreciate the reviewers' comments. We are pleased we were able to address their concerns regarding cofactor specificity with the additional experiments. Additionally, at this final stage, we have modified the manuscript to be in adherence of all *Cell Genomics* publication guidelines.